# The molecular basis of FimT-mediated DNA uptake during bacterial natural transformation

Sebastian A. G. Braus [1], Francesca L. Short [2], Stefanie Holz[1,3], Matthew J. M. Stedman[1,3], Alvar D. Gossert [1] & Manuela K. Hospenthal [1✉]

Naturally competent bacteria encode sophisticated protein machinery for the uptake and translocation of exogenous DNA into the cell. If this DNA is integrated into the bacterial genome, the bacterium is said to be naturally transformed. Most competent bacterial species utilise type IV pili for the initial DNA uptake step. These proteinaceous cell-surface structures are composed of thousands of pilus subunits (pilins), designated as major or minor according to their relative abundance in the pilus. Here, we show that the minor pilin FimT plays an important role in the natural transformation of *Legionella pneumophila*. We use NMR spectroscopy, in vitro DNA binding assays and in vivo transformation assays to understand the molecular basis of FimT's role in this process. FimT binds to DNA via an electropositive patch, rich in arginines, several of which are well-conserved and located in a conformationally flexible C-terminal tail. FimT orthologues from other *Gammaproteobacteria* share the ability to bind to DNA. Our results suggest that FimT plays an important role in DNA uptake in a wide range of competent species.

[1] Institute of Molecular Biology and Biophysics, ETH Zürich, Otto-Stern-Weg 5, Zürich, Switzerland. [2] Department of Microbiology, Monash University, 19 Innovation Walk, Clayton 3800 Victoria, Australia. [3] These authors contributed equally: Stefanie Holz, Matthew J. M. Stedman. ✉email: manuela.hospenthal@mol.biol.ethz.ch

Competent bacteria can take up exogenous DNA, present in their environment, and integrate it into their genomes by the process of natural transformation. This is an important avenue of horizontal gene transfer (HGT), which has widespread consequences for bacterial evolution and the spread of antibiotic resistance and other pathogenicity traits. In contrast to other modes of HGT, namely transduction and conjugation, natural transformation is entirely controlled by the recipient cell that encodes all the required machinery for DNA uptake, translocation and integration[1]. More than 80 bacterial species, including Gram-negative and Gram-positive organisms, have been shown to be naturally competent[2], yet the true prevalence of this mechanism amongst bacteria likely remains underappreciated. The Gram-negative bacterium *Legionella pneumophila* is naturally competent[3], consistent with the observation that its genome bears evidence of frequent HGT and recombination events[4–6]. Although *L. pneumophila* could be described as an accidental human pathogen, it is the aetiological agent of Legionnaire's disease, a serious and life-threatening form of pneumonia, that results from an infection of alveolar macrophages by contaminated aerosols[7,8].

*Legionella*, like most Gram-negative bacteria, are thought to utilise type IV pili (T4P) for DNA uptake[3,9,10], which is defined as the movement of DNA across the outer membrane (OM) and into the periplasmic space[11]. However, the molecular mechanisms involved in this step remain poorly defined. T4P are extracellular proteinaceous appendages composed of thousands of individual pilus subunits (pilins), designated as major or minor depending on their relative abundance in the pilus[12,13]. A prevailing model suggests that T4P can bind to DNA[9] and transport it into the cell via pilus retraction, which is powered by the retraction ATPase PilT[14,15]. Pilus retraction is thought to bring the DNA into proximity with the OM and be taken up across the OM-embedded secretin channel PilQ, which is the same pore traversed by the T4P themselves[16,17]. Once in the periplasm, ComEA binds to incoming DNA to prevent its back-diffusion by acting like a Brownian ratchet[18,19]. Subsequently, DNA is converted into single-stranded DNA (ssDNA) and transported across the inner membrane (IM) by a putative channel called ComEC[20]. In the cytoplasm, ssDNA is protected by single-stranded DNA binding protein (Ssb)[21] and DNA processing protein A (DprA)[22], before being integrated into the genome by homologous recombination in a RecA- and ComM-dependent manner[23,24].

In recent years, studies of several competent bacteria have shown that their T4P (or their pilins) can directly interact with DNA[15,25–29]. This function was attributed to specialised minor pilins or pilin-like proteins in *Neisseria species* (ComP)[27,28], *Vibrio cholerae* (VC0858 and VC0859)[15], and *Thermus thermophilus* (ComZ)[29], although a major pilin (PilA4) has also been suggested to contribute in the latter[30]. Of these, ComP found in *Neisseria* species, is the best-characterised DNA-binding minor pilin to date. ComP displays a sequence preference for neisserial DNA containing so-called DNA uptake sequences (DUS)[31–33] and binds to DNA through an electropositive surface patch[27,28,34]. VC0858, VC0859 and ComZ are thought to be located at the pilus tip[15,29], whereas ComP has been suggested to either be incorporated throughout the pilus fibre[28] or at the pilus tip[9]. In addition to these proteins, the minor pilin FimT has also been implicated in natural transformation, as its loss leads to a reduction in transformation efficiency in *Acinetobacter baylyi*[35]. However, this phenotype was never followed up with further DNA-binding studies.

We set out to study DNA uptake during natural transformation in *Legionella pneumophila*. It is not known whether Legionella's T4P can interact with DNA, and if so, which pilins are responsible. Here we show that FimT efficiently interacts with DNA in vitro and in vivo, and that loss of binding, just like *fimT* deletion, results in almost complete abrogation of natural transformation. We present the structure of FimT and show that a conserved electropositive surface patch rich in arginines is required for DNA binding. Finally, we show that FimT is not only important for natural transformation in *L. pneumophila*, but that it likely plays a role in many other bacterial species, as suggested by DNA binding studies and bioinformatic analyses. Together, our work provides the molecular basis of FimT's role in natural transformation.

## Results

**FimT is critical for natural transformation in *L. pneumophila* and interacts with DNA.** *L. pneumophila* encodes type IV pilins in several genetic loci. The largest of these operons contains *fimU*, *pilV*, *pilW*, *pilX*, *pilY1* and *pilE*, and is shared, in a similar arrangement, by many other T4P-containing organisms[36]. In addition, the *L. pneumophila* Philadelphia-1 genome encodes several other type IV pilins and putative type IV pilins in locations seemingly far removed from the *fimU*-containing operon, or indeed from other genes encoding components of the T4P machinery. Among these are two consecutively encoded PilA homologues, PilA1 and PilA2, three further putative pilin genes, *lpg1995*, *lpg1996* and *lpg1997*, arranged in an operonic structure, and *fimT*, which appears entirely on its own. Recent work has suggested that PilA2 is the major pilin in *L. pneumophila* and therefore required for natural transformation, while PilA1 appears dispensable[10].

FimT and FimU are minor type IV pilins that belong to the GspH/FimT family of proteins (Pfam: PF12019; InterPro: IPR022346), which also includes the type II secretion system (T2SS) pseudopilin GspH/XcpU. All three genes are encoded in the *L. pneumophila* genome and share an overall amino acid sequence identity of ~15–25%. *L. pneumophila* FimT (FimT$_{Lp}$) and FimU (FimU$_{Lp}$) possess all the features of typical type IV pilins, including an N-terminal signal sequence required for their targeting to the inner membrane (IM), followed by a hydrophobic transmembrane helix required for IM insertion prior to pilus assembly and proper packing into the filament structure post assembly[12,37]. To test whether FimT$_{Lp}$ or FimU$_{Lp}$ play a role in natural transformation in *L. pneumophila*, we performed transformation assays comparing the *fimT* and *fimU* deletion strains with the parental strain and strains harbouring deletions in genes known to be important for natural transformation (Fig. 1a). Deletion of *comEC*, encoding the putative IM DNA channel, *pilQ*, encoding the OM secretin, and *pilT*, encoding the retraction ATPase, resulted in undetectable levels of natural transformation in our assay. These observations are in close agreement with previous studies in *L. pneumophila*, as well as other competent Gram-negative organisms such as *V. cholerae*, where deletion of these genes resulted in a complete loss or severe reduction of natural transformation[10,16]. Deletion of *fimU* did not produce a phenotype, whereas natural transformation was undetectable in the *fimT* deletion strain, as observed previously in *A. baylyi*[35]. Expression of FimT$_{Lp}$ in trans from an IPTG-inducible promoter restored the transformation efficiency of our *L. pneumophila* strain to wild-type levels, showing that the transformation defect is specific to FimT$_{Lp}$. Although the absence of FimT does not appear to affect pilus biogenesis or other T4P functions such as twitching motility in other organisms[35,38,39], this possibility cannot be ruled out.

We reasoned that FimT contributes to the OM DNA uptake step of natural transformation by forming a constituent part of type IV pili (T4P) able to directly bind to DNA. Therefore, we performed electrophoretic mobility shift assays (EMSA) to test

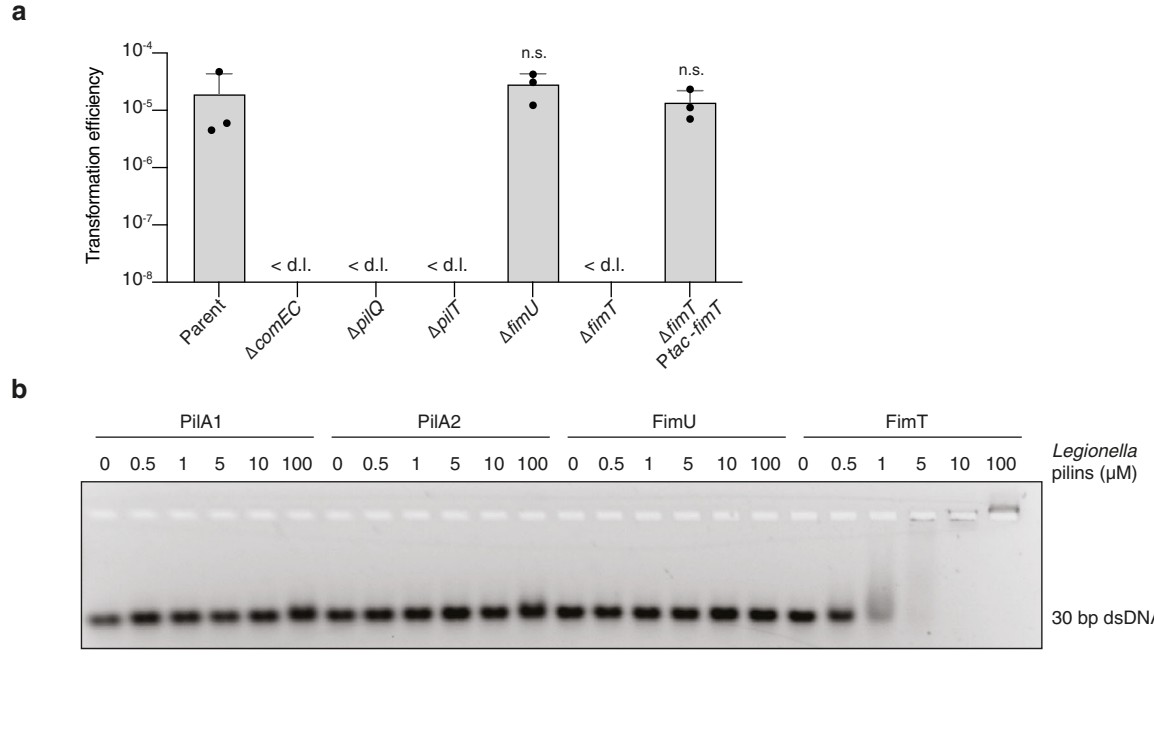

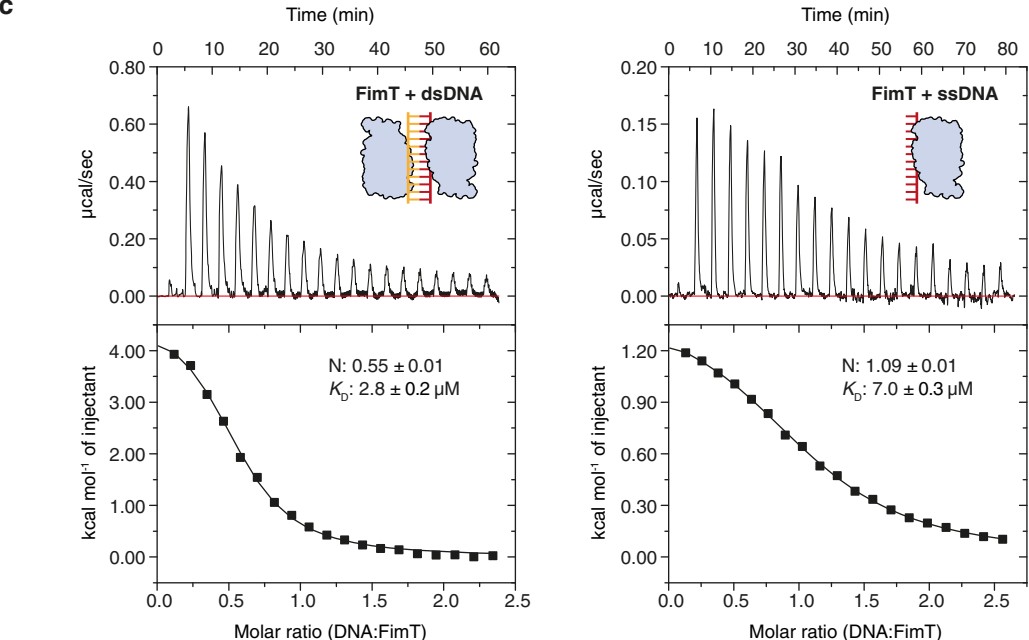

**Fig. 1 FimT is critical for the transformation of *L. pneumophila* and binds to DNA. a** Natural transformation efficiencies of the parental *L. pneumophila* Lp02 strain and Lp02 strains harbouring deletions of genes known to play a role in transformation compared to the *fimU* and *fimT* deletion strains. The Δ*fimT* strain was complemented by ectopic expression of wild-type FimT, under the control of an IPTG-inducible promoter. The mean transformation efficiencies of three independent biological replicates is shown (error bars represent standard deviation [SD]). < d.l., below detection limit (d.l.) (average d.l. = $2.0 \times 10^{-8} \pm 8.2 \times 10^{-9}$). Statistical significances of transformation differences were determined on log-transformed[83] data using an unpaired two-sided t-test with Welch's correction. All strains were compared to the parental strain. n.s., not statistically significant, $p > 0.05$ ($p_{\Delta fimU} = 0.39$; $p_{\Delta fimT\ Ptac-fimT} = 0.89$). **b** In vitro DNA binding of purified *L. pneumophila* PilA1, PilA2, FimU and FimT assessed by an EMSA. A 30 bp dsDNA fragment (1 μM) was incubated with increasing concentrations of purified pilins (0–100 μM) and resolved by agarose gel electrophoresis. This experiment was independently performed three times with reproducible results. **c** ITC binding studies of wild-type FimT binding to 12meric dsDNA (left) and ssDNA (right). In both cases, DNA (syringe) was injected into FimT (cell). Data were fitted using the "one set" of sites model, assuming that both binding sites on the dsDNA are of equal affinity. Source data are provided as a Source Data file.

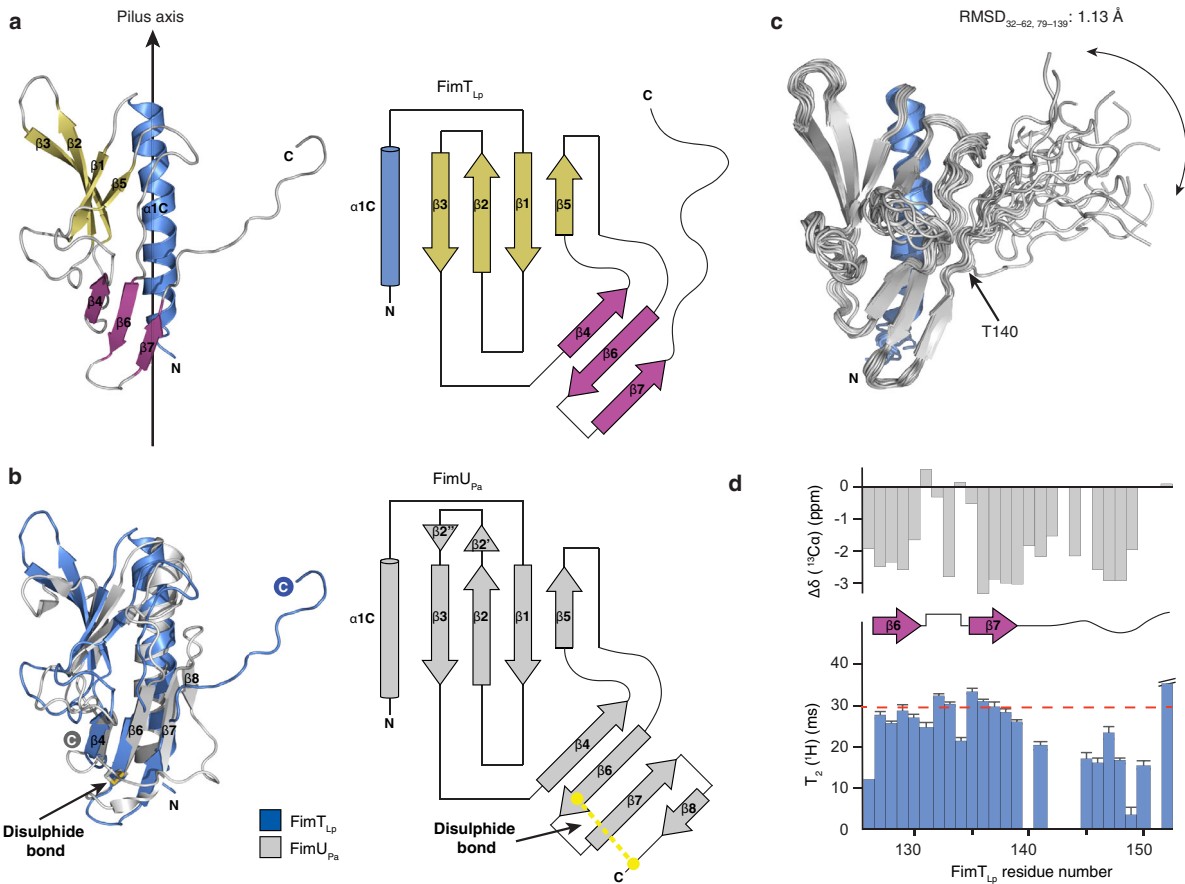

**Fig. 2 The structure of FimT$_{Lp}$. a** The solution structure of FimT$_{Lp}$ 28–152 (state 18) in ribbon representation (left) and the corresponding topology diagram (right). Secondary structure elements are indicated: truncated N-terminal α-helix (α1C) (blue), β-sheet I formed by β1, β2, β3, and β5 (yellow), and β-sheet II formed by β4, β6, and β7 (magenta). A vertical arrow indicates the pilus axis from the cell surface towards the pilus tip. **b** Structure alignment of FimT$_{Lp}$ (blue) and FimU$_{Pa}$ (grey; PDB ID: 4IPV) (left) and the topology diagram of FimU$_{Pa}$ (right). The disulphide bond of FimU$_{Pa}$ is indicated in stick representation with sulphur atoms in yellow. **c** Superimposed 20 lowest energy structures calculated by NMR spectroscopy. An arrow indicates the conformational flexibility of the C-terminal tail (140–152). The pairwise backbone root-mean-square deviation (RMSD) for the structured region (residues 32–62, 70–139) is 1.13 Å. N- and C-termini are indicated in each panel. **d** Cα chemical shift values (top) and T$_2$($^1$H) transverse relaxation data (bottom), encompassing the last 27 residues of FimT$_{Lp}$. Secondary structural elements are indicated and error bars represent the fitting errors of the respective exponential decay curves. Source data are provided as a Source Data file.

whether FimT$_{Lp}$ is able to bind to DNA in vitro (Fig. 1b). In order to produce soluble protein samples, all pilins were expressed as truncations lacking the N-terminal transmembrane helix (Supplementary Fig. 1a). Indeed, purified FimT$_{Lp}$ interacted with all DNA probes tested, including ssDNA, dsDNA, linear and circular DNA molecules, whereas neither FimU$_{Lp}$, nor the putative major pilin subunits (PilA1 and PilA2) showed any interaction (Fig. 1b and Supplementary Fig. 1b). These experiments suggest that the dissociation constant ($K_D$) of the interaction between FimT$_{Lp}$ and 30meric DNA is in the low μM range. In order to determine the $K_D$ more precisely and to learn about the binding stoichiometry of this interaction, we performed isothermal titration calorimetry (ITC) utilising shorter 12meric ssDNA or dsDNA fragments (Fig. 1c). We determined a $K_D$ of 7.0 μM and 2.8 μM for 12meric ssDNA and dsDNA, respectively. Interestingly, these experiments revealed that a single FimT$_{Lp}$ binds to 12meric ssDNA, whereas two molecules can bind to the dsDNA ligand, suggesting two binding sites on opposite sides of the double helix.

**The solution structure of FimT$_{Lp}$.** We determined the solution structure of the soluble N-terminally truncated (residues 28–152,

mature pilin sequence numbering) FimT$_{Lp}$ by nuclear magnetic resonance (NMR) spectroscopy (Fig. 2a, Supplementary Table 1). The structure consists of an N-terminal α-helix (α1 C) (the transmembrane portion of this helix, α1 N, has been removed in the construct), two β-sheets that complete the C-terminal globular pilin domain, and a C-terminal tail, which exhibits conformational flexibility. Both β-sheets are composed of antiparallel strands: β-sheet I is formed by β1, β2, β3 and β5, and β-sheet II by β4, β6 and β7. The closest structural homologue is FimU from *Pseudomonas aeruginosa* (FimU$_{Pa}$) (PDB ID: 4IPU, 4IPV) (Fig. 2b). While the two structures share a common fold, there are some key differences. In the FimU$_{Pa}$ structure, the loop between β2 and β3 in β-sheet I forms an additional β-hairpin (β2′ and β2″). It is possible, however, that this additional β-hairpin of FimU$_{Pa}$ simply represents the conformation captured in the crystal structure, as the length of the β2-β3 loop is similar in both proteins. In addition, the β7 strand of β-sheet II is longer in FimU$_{Pa}$, and FimU$_{Pa}$ contains an additional strand (β8)[36]. Furthermore, FimU$_{Pa}$ contains a disulphide bond connecting Cys127 of β6 to the penultimate residue, Cys158, effectively stapling the C-terminal tail in place on top of β-sheet II. Such a disulphide bond is found in various major and minor pilins and the

intervening sequence is known as the D-region[37,40]. Further structures of GspH/FimT family proteins exist, including of the minor T2SS pseudopilins, GspH from *Escherichia coli* (PDB ID: 2KNQ) and its orthologue EpsH from *V. cholerae* (PDB ID: 2QV8[41] and 4DQ9[42]), which display similar folds (Supplementary Fig. 2).

The C-terminal tail (residues 140–152) of $FimT_{Lp}$ is unique amongst the currently determined structures of GspH/FimT family members. Different pieces of NMR data suggest significant conformational exchange, but not an entirely flexibly disordered tail. The amide resonances of residues 140–149 are very weak and those of residues 142–144 are not visible at all. We could not observe any intense long-range nuclear Overhauser effects (NOEs) for residues 140–152, which would be expected for a well-defined β-sheet conformation. $T_2$ relaxation measurements indicated conformational exchange on the millisecond timescale, as the $T_2$ values for backbone amide $^1H$ and $^{15}N$ nuclei for the C-terminal tail were approximately half the value of the structured part of the protein (Fig. 2d, Supplementary Fig. 3). A fully disordered C-terminal tail could however be excluded by {$^1H$}-$^{15}N$ heteronuclear NOE measurements, as the NOE intensity for the amides in the tail was close to the theoretical value of 0.78, which is expected for amides on globular particles. Finally, the deviations of Cα chemical shifts from random coil values clearly indicated a β-strand propensity (Fig. 2d, Supplementary Fig. 3). The data therefore suggest that the C-terminal amino acids have a β-strand-like backbone conformation but sample different states in the micro- to millisecond timescale. These findings are further supported by low amide proton temperature coefficients[43] and increased proteolytic susceptibility of this region, compared to the rest of the structure, witnessed by disappearance of the NMR resonances of the tail after prolonged storage of samples.

**$FimT_{Lp}$ interacts with DNA via a conserved C-terminal region rich in arginines**. Next, we characterised the residues of $FimT_{Lp}$ involved in DNA binding using NMR spectroscopy (Fig. 3a–c). We performed binding experiments titrating increasing amounts of 12 bp dsDNA into $^{15}N$-labelled $FimT_{Lp}$ and recorded $^{15}N$, $^1H$ heteronuclear single-quantum correlation (2D [$^{15}N,^1H$]-HSQC) spectra. Most $FimT_{Lp}$ resonances remained unperturbed (Fig. 3a), which suggests that no global conformational change occurs upon DNA binding. However, a subset of resonances exhibit marked chemical shift perturbations (CSPs) (Fig. 3a), indicating changes in the local chemical environment resulting from direct contact with DNA or other indirect conformational changes. A plot of CSPs against the amino acid sequence is shown in Fig. 3b, and we mapped CSPs greater than a threshold (Δppm > 1σ) onto the $FimT_{Lp}$ surface (Fig. 3c, Supplementary Fig. 4). The largest CSPs correspond to residues located in three adjacent loop regions in the C-terminal globular domain of the protein, the β4-β5 loop (residues 103–106), the β5-β6 loop (118–126) and the C-terminal tail (140–152) (Fig. 3b). These shifts predominantly map to an elongated surface patch connecting the C-terminal tail with the globular C-terminal domain of $FimT_{Lp}$ (Fig. 3c). Most of these residues are predicted to be accessible in the context of the assembled pilus, particularly when considering the flexibility of this region (Fig. 2d, Supplementary Fig. 3). CSPs corresponding to residues outside this contiguous surface patch can be explained by indirect conformational changes. We attempted to further structurally characterise the DNA-bound state, with special emphasis on possible changes in the structure or dynamics of the C-terminus. However, the $FimT_{Lp}$-DNA complex was not stable long-term and NMR signals were generally strongly weakened upon DNA binding, such that relaxation or triple resonance experiments did not yield spectra of sufficient quality. An

analysis of evolutionary conservation of the $FimT_{Lp}$ surface revealed that many of the interacting residues are also well conserved (Fig. 3c). In particular, residues of the C-terminal tail show marked sequence conservation and include a number of positively charged arginines, which are often involved in protein-DNA contacts through binding to the negatively charged DNA backbone *via* electrostatic interactions[44].

**Interface mutations inhibit DNA binding and natural transformation in vivo**. We conducted microscale thermophoresis/temperature-related intensity change (MST/TRIC) experiments to measure the binding of labelled 12 bp dsDNA to purified $FimT_{Lp}$ variants (Supplementary Fig. 1a), in order to further understand the nature of the $FimT_{Lp}$-DNA interaction and the importance of specific interface residues. First, we conducted experiments under different buffer conditions to test whether the affinity of the interaction between wild-type $FimT_{Lp}$ and DNA is dependent on ionic strength. Indeed, when we increased the NaCl concentration from 50 mM to 150 mM, thereby raising the ionic strength, the $K_D$ increased from ~6.3 μM to ~70.1 μM (Fig. 4a). This is consistent with a non-sequence specific protein-DNA interaction, which is electrostatically driven. Furthermore, the $K_D$ determined at a NaCl concentration of 50 mM agrees very well with the affinities determined from the ITC experiments ($K_D$ of 2.8 μM) (Fig. 1c), as well as our NMR binding studies ($K_D$ of ~8.1 μM) (Supplementary Fig. 5), which were all conducted in the same buffer. Next, we used MST/TRIC to test the importance of several charged residues at the DNA binding interface identified by our NMR analyses (Fig. 4b). We substituted arginine or lysine residues for glutamine in the three loop regions we identified to be important for binding. As expected, the loss of a single charged residue (e.g., K103 in the β4-β5 loop; R119 in the β5-β6 loop; R143, R146 or R148 in the C-terminal tail) only led to a small reduction in the affinity (~1.4–4 fold). However, the combined loss of two (R146/R148) or three (R143/R146/R148) charged residues next to each other on the $FimT_{Lp}$ surface was more detrimental to binding, resulting in a ~10 fold or greater reduction in affinity.

In the absence of FimT, Lp02 cells are not transformable (Fig. 1a). However, we wanted to rule out that this phenotype could result as a consequence of a pilus biogenesis defect by testing what effect our DNA binding mutations have on natural transformation in vivo (Fig. 4c). We reasoned that our substitution mutations are unlikely to affect pilus biogenesis given that they are located in a portion of the pilin not known to play a role in pilus assembly or packing. Additionally, we generated control mutations in residues of FimT not involved in DNA binding, within the same three regions (β4-β5 loop, β5-β6 loop and the C-terminal tail) previously identified to contribute to the DNA binding patch of FimT (Fig. 4d). These data show that mutations of the DNA binding residues of FimT lead to reduction or loss of transformation, whereas our FimT control mutations in residues not involved in DNA binding support wild-type levels of transformation in vivo. Mutation of a single charged residue involved in DNA binding reduces *Legionella's* transformability by ~30–600 fold, whereas the double and triple mutants completely abrogate DNA uptake in our assay and thus phenocopy the effect observed upon *fimT* deletion (Fig. 1a). These results suggest that our FimT mutants are incorporated into the pilus fibre and that the reduction or loss of transformation efficiency is due to the diminished ability of the FimT mutants to bind to DNA in the context of a DNA uptake pilus.

**FimT of other Gram-negative bacteria also interacts with DNA.** Given that FimT, and the residues involved in DNA binding

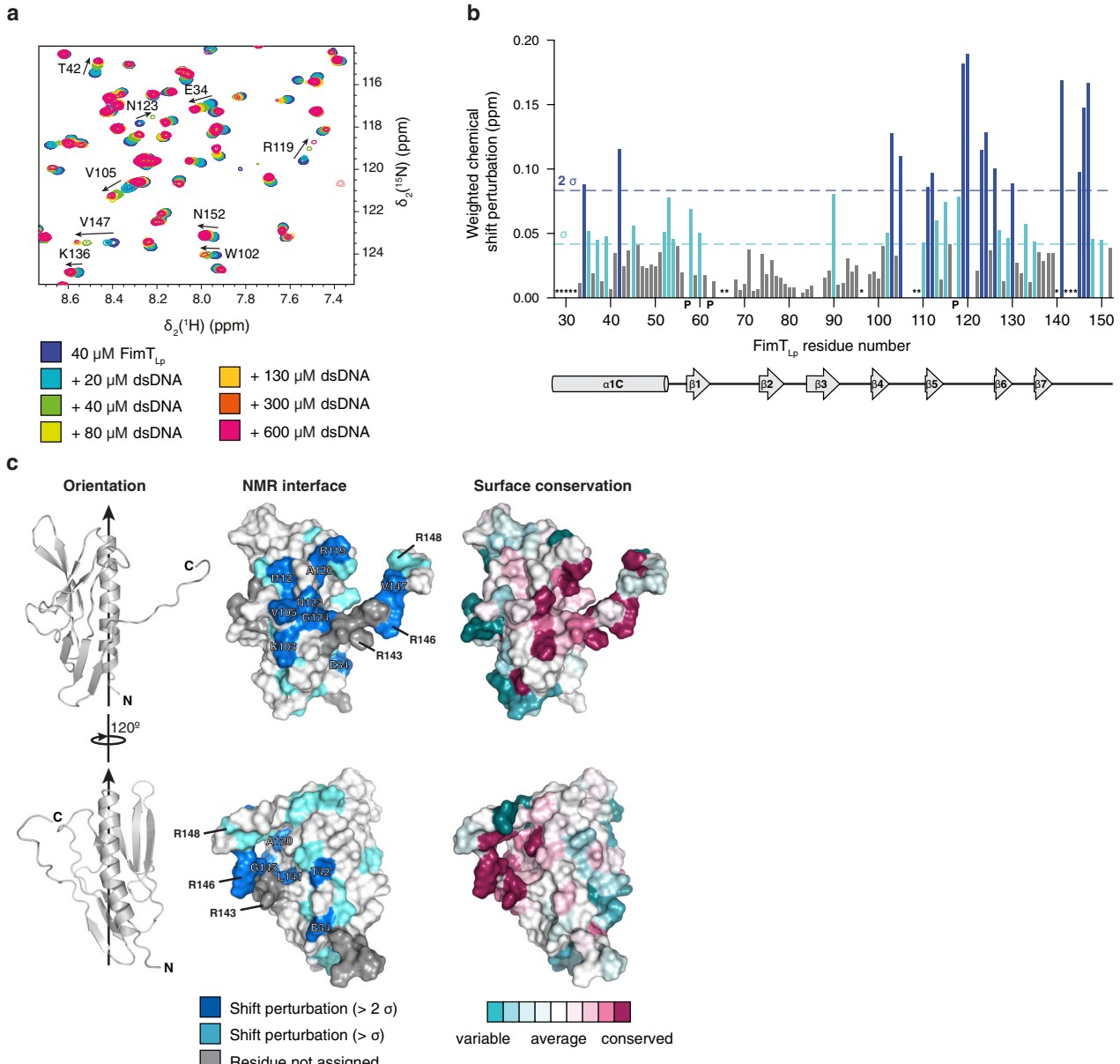

**Fig. 3 Identification of the DNA interaction surface of FimT$_{Lp}$. a**, Selected region of $^1$H, $^{15}$N-HSQC spectra showing $^{15}$N-labelled FimT$_{Lp}$ alone and in the presence of increasing concentrations of 12 bp dsDNA. **b**, Weighted CSP map generated from a. Residues experiencing CSPs ($\Delta$ppm > 1 $\sigma$), light blue; residues experiencing CSPs ($\Delta$ppm > 2 $\sigma$), dark blue; P, prolines; *, residues not assigned. **c**, Left, FimT$_{Lp}$ is shown in two orientations rotated by 120° in ribbon representation. Arrows indicate the pilus axis as in Fig. 2a. Middle, CSPs are mapped onto the surface of FimT$_{Lp}$ and coloured as in b. Residues producing large shifts are labelled on the molecular surface. Right, surface residues of FimT$_{Lp}$ are coloured according to conservation. This image was generated using the ConSurf server[84]. Source data are provided as a Source Data file.

identified in FimT$_{Lp}$, appear to be conserved, we wondered whether FimT orthologues from other bacteria are also capable of binding DNA. We expressed and purified FimT and FimU from the human pathogen *P. aeruginosa* and the plant pathogen *Xanthomonas campestris* (both γ-Proteobacteria) and performed EMSAs to assess DNA binding in vitro (Fig. 5a). Interestingly, FimT from both species binds to DNA and the affinity appears to be within the same order of magnitude as *L. pneumophila* FimT. On the other hand, FimU does not interact with DNA, except for the *X. campestris* homologue, which shows very weak binding at very high FimU concentrations. This may be due to FimU$_{Xc}$'s slightly more basic C-terminal tail region, and/or other the presence of other residues elsewhere in the protein that allow FimU$_{Xc}$ to weakly interact with DNA.

Since both FimT orthologues (FimT$_{Pa}$ and FimT$_{Xc}$) likely share structural similarities to FimT$_{Lp}$, we tested whether they are capable of restoring natural transformability in a *L. pneumophila fimT* deletion strain. First, we ectopically expressed full-length FimT$_{Pa}$ and FimT$_{Xc}$ orthologues and performed a transformation assay. In this set-up, natural transformation was not restored in the *Legionella* system (Fig. 5b). Possible reasons for this lack of complementation include that the signal sequences of the orthologues are not compatible with the *Legionella* system and do not lead to IM localisation, that the orthologues do not interact appropriately with the *Legionella* T4P machinery during assembly, that the orthologues' N-terminal helices do not allow for proper incorporation into or packing with the *Legionella* T4P filament, and that the orthologues do not promote DNA binding

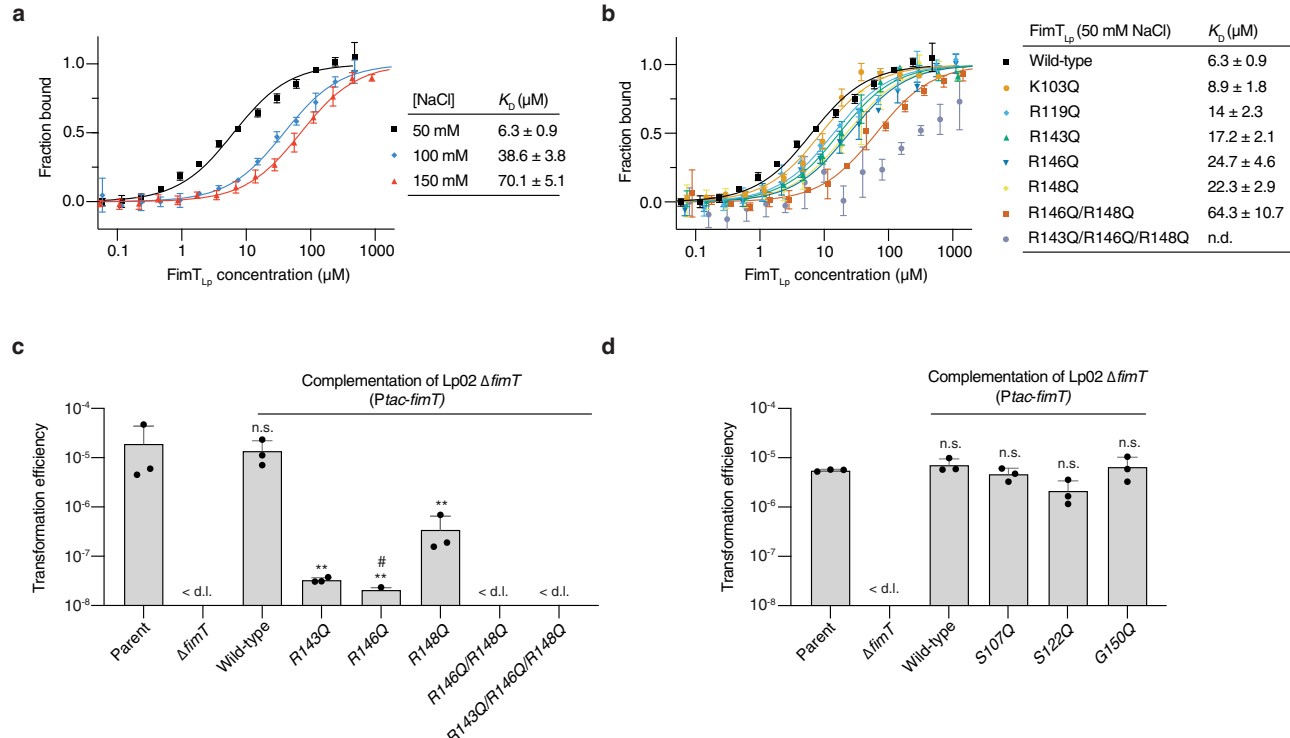

**Fig. 4 Characterisation of FimT$_{Lp}$ binding to DNA in vitro and in vivo.** MST/TRIC binding assay of 12 bp FAM-labelled dsDNA with **a**, wild-type FimT$_{Lp}$ performed at different NaCl concentrations (ionic strength) and **b**, wild-type FimT$_{Lp}$ compared to FimT mutants predicted to disrupt DNA binding based on Fig. 3. n.d., not determined. The MST/TRIC data were fitted according to two binding sites with equal affinity. Error bars represent the mean ± SD (n = 3). **c**, **d**, Natural transformation efficiencies of parental Lp02, Lp02 Δ*fimT*, and the Lp02 Δ*fimT* strain complemented by ectopic expression of wild-type and FimT$_{Lp}$ mutants that disrupt DNA binding (**c**) or do not contribute to DNA binding according to Fig. 3 (**d**). The mean transformation efficiencies of three independent biological replicates are plotted with error bars representing the SD. < d.l., below detection limit (d.l.) (average d.l. = $2.0 \times 10^{-8} \pm 8.2 \times 10^{-9}$ (c) and $2.5 \times 10^{-8} \pm 9.5 \times 10^{-9}$ (d)); #, below d.l. in at least one replicate (average d.l. used to calculate the mean transformation efficiency). The assay in panel c was performed in parallel to those displayed in Fig. 1a, and statistical differences were determined on log-transformed data using an unpaired two-sided t-test with Welch's correction. The Lp02 Δ*fimT* strain complemented with mutants was compared to the Lp02 Δ*fimT* strain complemented with wild-type FimT, which was in turn compared to the parental strain. **, $p < 0.01$ ($p_{R143Q} = 0.003$; $p_{R146Q} = 0.002$; $p_{R148Q} = 0.004$); n.s., not statistically significant, $p > 0.05$ ($p_{Wild-type(c)} = 0.89$; $p_{Wild-type(d)} = 0.33$; $p_{S107Q} = 0.38$; $p_{S122Q} = 0.08$; $p_{G150Q} = 0.89$). Source data are provided as a Source Data file.

in vivo. To circumvent possible limitations of targeting and assembly inherent with introducing a non-native protein into another system, we ectopically expressed chimeric proteins, where we replaced only the flexible C-terminal tail region (lacking a disulphide bond) of FimT$_{Lp}$ with the bona fide D-region of the FimT orthologues. Intriguingly, the chimeric construct, appending the D-region of FimT$_{Pa}$ to the rest of FimT$_{Lp}$, restored natural transformation to near wild-type levels (Fig. 5b). This suggests that when the parts of the pilin known to be critical for proper pilus biogenesis are native to the system, the D-region from another FimT orthologue is able to restore functionality in vivo. Together, these results indicate that DNA binding by FimT is not unique to *L. pneumophila* and that FimT may be important for DNA uptake in a wide range of competent species.

We then used genomic context and sequence information from the four FimT orthologs known to either bind to DNA or contribute to competence (from *L. pneumophila*, *X. campestris*, *P. aeruginosa* and *A. baylyi*) to explore the distribution and conservation of this protein (see Methods). First, we looked at the genetic location and organisation of FimT and FimU in *Legionella* and other bacteria (Supplementary Fig. 6). In *L. pneumophila*, *fimU* (*lpg0632*) is encoded in a minor pilin operon upstream of *pilV* (*lpg0631*), *pilW* (*lpg0630*), *pilX* (*lpg0629*), *pilY1* (*lpg0628*) and *pilE* (*lpg0627*). In contrast, *fimT* (*lpg1428*) appears as an 'orphan' gene, encoded elsewhere in the genome, and seemingly distant from genes encoding other type IV pilins, components of the T4P machinery

or genes with known functions in natural transformation. Interestingly, while *fimT* in other species could be found either as an orphan, or adjacent to other minor pilin-related genes, the location of *fimU* was conserved, and this pattern was seen in a broader collection of homologues as well as the functionally characterised representatives. We then retrieved a diverse set of homologues of FimT$_{Lp}$ and classified them according to genomic location and sequence similarity to exclude sequences that were likely to be FimU proteins. We found that FimT is conserved in all sequenced *Legionella* species, and homologues are found in a wide variety of γ-Proteobacteria from various phylogenetic orders, with representatives of the Xanthomonadales, Alteromonadales and Pseudomonadales being particularly common (Fig. 5c). The pairwise sequence identity was 40–50% between FimTs from *Legionella pneumophila* and other *Legionella* species, and ~25% (median) between *L. pneumophila* FimT and those from more distantly related bacteria. Around half of the FimT homologues are located in proximity (within 5 kb) to other minor pilin locus components. FimU is also present in many bacterial species, albeit not all species encode both genes. Phylogenetic analysis of FimT homologues showed that these proteins largely clustered with others from the same order and sharing the same locus type, indicating that *fimT* is likely to be vertically inherited. The best conserved regions of FimT include the N-terminal helix, important for pilus biogenesis (IM insertion, assembly, and structural packing), and the C-terminal region (Fig. 5d). This region of conservation includes many of the residues

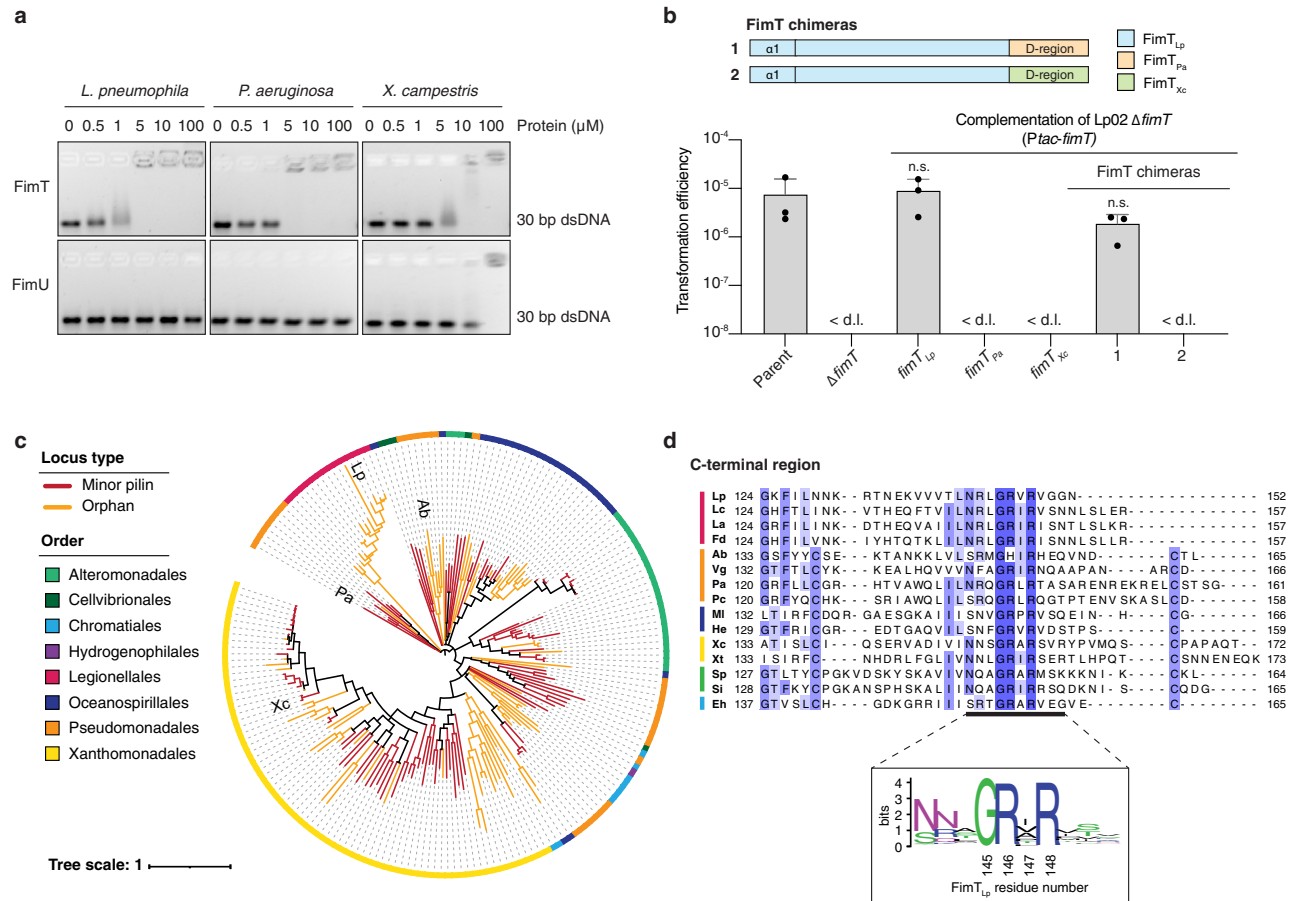

**Fig. 5 Bioinformatic and functional analysis of FimT orthologues. a** EMSA showing in vitro DNA binding of purified FimT and FimU orthologues from *L. pneumophila*, *P. aeruginosa* and *X. campestris*. A 30 bp dsDNA fragment (1 μM) was incubated with increasing concentrations of purified pilins (0–100 μM) and resolved by agarose gel electrophoresis. These experiments were independently performed three times with reproducible results. **b** A comparison of natural transformation efficiencies of the Lp02 Δ*fimT* strain complemented by ectopic expression of FimT$_{Lp}$, FimT orthologues from *P. aeruginosa* (FimT$_{Pa}$) and *X. campestris* (FimT$_{Xc}$), or chimeric FimT mutants (1–2). The corresponding composition of these FimT chimeras (1–2) is explained by a schematic drawing (top). The mean transformation efficiencies of three independent biological replicates are shown with error bars representing the SD. < d.l., below detection limit (d.l.) (average d.l. = $4.8 \times 10^{-8} \pm 2.1 \times 10^{-8}$). An unpaired two-sided t-test with Welch's correction, using log-transformed data, was used to analyse statistical significance. The Lp02 Δ*fimT* strain complemented with the chimeric construct was compared to the Lp02 Δ*fimT* strain complemented with wild-type FimT, which was in turn compared to the parental strain. n.s., not statistically significant, $p > 0.05$ ($p_{fimTLp} = 0.69$; $p_1 = 0.1$). **c** Phylogenetic tree of FimT homologues, comprising eight orders of γ-Proteobacteria illustrated by the coloured circumferential ring. Branches coloured in orange represent FimTs encoded as orphan genes, whereas those coloured red represent FimTs encoded within minor pilin operons. The positions of the four functionally characterised FimT orthologues in the tree are indicated (Lp, *L. pneumophila*; Ab, *A. baylyi*; Pa, *P. aeruginosa*; and Xc, *X. campestris*). The scale bar indicates the average number of substitutions per site. **d** Top, multisequence alignment of representative FimT orthologues across six orders (indicated by a coloured line as in c) focusing on their C-terminal region (Lc, *Legionella cherrii*; La, *Legionella anisa*; Fd, *Fluoribacter dumoffii*; Vg, *Ventosimonas gracilis*; Pc, *Pseudomonas chloritidismutans*; Ml, *Marinicella litoralis*; He, *Halomonas endophytica*; Xt, *Xylella taiwanensis*; Sp, *Shewanella polaris*; Si, *Shewanella indica*; Eh, *Ectothiorhodospira haloalkaliphile*). Residues are coloured according to sequence identity. Bottom, sequence logo generated from the full multisequence alignment of 196 high-confidence FimTs. Source data are provided as a Source Data file.

we have identified to be important for DNA binding and thus natural transformation (Fig. 3c, d). Indeed, it appears as though these DNA binding residues can be identified in proteins with as little as 18% overall amino acid sequence identity with FimT$_{Lp}$. Taken together, FimT homologues share an overall fold and a conserved DNA-binding motif near the C-terminus of the protein, and can be found in diverse genomic locations within diverse proteobacterial species.

## Discussion

Natural transformation is an important mode of horizontal gene transfer with widespread consequences for bacterial evolution. Furthermore, the spread of pathogenicity traits and antibiotic resistance genes leads to the emergence of increasingly virulent and difficult to treat bacterial strains. The first step of this process involves DNA uptake mediated by T4P[9], which has only been studied in a handful of competent species. The minor type IV pilin FimT, but not the closely related FimU, from *A. baylyi* was previously implicated in natural transformation[35], yet its mechanism remained obscure. Here, we characterised FimT from the naturally competent human pathogen *L. pneumophila* (FimT$_{Lp}$) and revealed the molecular mechanisms underlying its role in natural transformation.

We hypothesised that FimT$_{Lp}$ is involved in DNA uptake by binding to extracellular DNA in the context of T4P and showed that *Legionella* strains lacking *fimT* display a marked reduction in transformation efficiency (Fig. 1a). Indeed, purified FimT$_{Lp}$

interacted with DNA in vitro, regardless of the nature of DNA probe tested (Fig. 1b, Supplementary Fig. 1b). Furthermore, we determined the structure of $FimT_{Lp}$ by NMR spectroscopy (Fig. 2) and mapped its DNA interaction surface by chemical shift perturbation experiments (Fig. 3). This binding surface consists of several positively charged residues, some of which are highly conserved, located primarily in two loop regions (the β4-β5 and β5-β6 loops) and the C-terminal tail (Fig. 3b, c). The importance of key residues for DNA binding and natural transformation was confirmed by in vitro DNA binding assays and in vivo transformation assays (Fig. 4b, c). Although our ITC experiments (Fig. 1c) indicate a 2:1 ($FimT_{Lp}$:dsDNA) binding mode in which two FimT molecules engage the DNA double helix from opposite sides, we do not think this is physiologically relevant in the context of the T4P. This is because at least two FimT moieties would need to be located near each other in the pilus and oriented such that DNA could be sandwiched between them, which would be difficult to achieve given that FimT's N-terminal helices would need to pack into the centre of the filament. Although it remains a possibility that more than one FimT, or indeed more than one DNA-binding pilin, are present within the T4P, producing an avidity effect.

Our structure of $FimT_{Lp}$ shares the same overall fold as the closely related T4P minor pilin $FimU_{Pa}$, and the T2SS minor pseudopilins $GspH_{Ec}$ and $EpsH_{Vc}$, albeit with some key differences (Fig. 2, Supplementary Fig. 2). In place of the last β-strand (β8), part of β-sheet II in all other currently determined FimT/GspH family structures, $FimT_{Lp}$ contains a conformationally flexible C-terminal tail (Supplementary Fig. 3). In our NMR studies, the heteronuclear {¹H}-¹⁵N NOE data and Cα chemical shifts for the C-terminal residues are indicative of a β-strand conformation, while the $T_2$ transverse relaxation times for backbone amide ¹H and ¹⁵N nuclei, increased line broadening and the absence of H-bonds indicate a less well-structured conformation. A plausible interpretation of these results is that this region can exchange between a β-strand and a less-structured conformation on a millisecond timescale. The flexibility of this region is further supported by its increased proteolytic susceptibility. $FimT_{Lp}$, as well as all FimT homologues from the order Legionellales, lack the D-region defining disulphide bond present in many major and minor pilins, including other FimT and FimU homologues (Fig. 5d). Therefore, it is likely that disulphide bond-containing FimT orthologues do not possess a conformationally flexible C-terminal tail. It remains to be seen whether or not the C-terminal tail of $FimT_{Lp}$ adopts a more structured conformation in complex with DNA and/or in the context of a fully assembled pilus. The uniqueness of $FimT_{Lp}$'s C-terminal tail is further highlighted by the observation that the structure of $GspH_{Ec}$ was also determined in solution by NMR spectroscopy, yet it possesses a clearly defined and complete four-stranded β-sheet II region. This suggests that this region, also shared by $FimU_{Pa}$ and $EpsH_{Vc}$, is not simply a result of crystal lattice effects (Fig. 2, Supplementary Fig. 2).

FimU and GspH have been suggested to serve as adaptors linking the tip subunits to the remainder of the filament structure[36,45–47]. Whereas minor pilins in general have been suggested to play a role in pilus priming/pilus biogenesis[36,47], the deletion of FimU, but not FimT affected pilus biogenesis in *P. aeruginosa* and *Pseudomonas syringae*[38,39]. Furthermore, FimU, but not FimT of *P. aeruginosa* has been shown to play a role in bacterial twitching motility[48]. In *A. baylyi* on the other hand, loss of either protein showed near wild-type levels of twitching, but FimT appeared to play a role in natural transformation[35]. Orthologues of the GspH pseudopilin are critical components of the T2SS and may play a role in binding to T2SS protein substrates[49]. To this end, the crystal structure of the *V. cholerae*

orthologue EpsH revealed an extended and disordered β4-β5 loop (Supplementary Fig. 2d), which has been proposed to play a role in substrate binding[42]. Interestingly, we have identified this same loop to contribute to $FimT_{Lp}$-DNA binding (Fig. 3b). Therefore it appears that, although sharing a common evolutionary origin[50], FimT/GspH family proteins have become functionally diverged and specialised for the binding of different macromolecular substrates[46,51]. In the case of $FimT_{Lp}$, a surface patch rich in arginines enables it to function in DNA uptake during natural transformation.

The currently best-characterised DNA binding minor pilin is ComP[27,28,34]. While ComP homologues seem to be restricted to species of the family *Neisseriaceae*[27], FimT homologues are present in diverse γ-Proteobacteria and some Hydrophilales (Fig. 5c). Both proteins share a conserved type IV pilin core structure, including the N-terminal helix and a four-stranded antiparallel β-sheet, but differ substantially in their C-terminal regions. In the case of ComP, this region is characterised by its so-called DD-region containing two disulphide bonds[27] (Supplementary Fig. 7). By contrast, FimT contains a second three-stranded antiparallel β-sheet followed by its conformationally flexible C-terminal tail and contains no disulphide bonds. In both proteins, important DNA binding residues are located near the C-terminus, which would be exposed to the solvent in the context of a fully assembled pilus[28]. Interestingly, competent *Neisseriaceae* species preferentially take up DNA sequences from related species[31–33]. This has been attributed to ComP's increased binding affinity towards DUS-sequences, which are DNA sequences that are highly enriched in their own genomes[27]. It was proposed that ComP engages DNA via an initial electrostatic attraction, followed by ComP's α1-β1, β1-β2, DD-region binding to successive grooves of the dsDNA to achieve specificity[28]. In contrast, no sequence selectivity has been reported for *L. pneumophila*[3], which is consistent with the electrostatic binding mode of $FimT_{Lp}$. In addition to $FimT_{Lp}$ and ComP, other type IV pilins or pilin-like proteins that contribute to T4P DNA binding include ComZ and PilA4 from *T. thermophilus*[29,30] and VC0858 and VC0859 from *V. cholerae*[15]. Once again, positively charged lysine and/or arginine residues likely contribute to DNA binding in all these proteins. Whereas VC0858, VC0859 and ComZ have been proposed to be located at the pilus tip[15,29], ComP was initially modelled within the pilus fibre[28], and later also suggested to potentially occupy a position within the pilus tip complex[9]. In the case of FimT, the precise localisation in the pilus remains to be determined, yet it seems plausible that it may be located within the pilus tip complex, given that its closest structural homologues (FimU, GspH and EpsH) are also proposed to occupy tip complex positions[45–47].

Lastly, we showed that other FimT orthologues, including FimT of the human pathogen *P. aeruginosa* and the plant pathogen *X. campestris*, are also capable of DNA binding (Fig. 5a). These experiments showed that FimT orthologues, whether they contain or lack the D-region defining disulphide bond, are capable of DNA binding. This was demonstrated even more strikingly by the FimT chimera, where the fusion of $FimT_{Lp}$ with $FimT_{Pa}$ introduced a non-native disulphide bond into the *Legionella* system, yet resulted in a functional protein in vivo capable of supporting natural transformation (Fig. 5b). This observation further supports the notion that perturbation of FimT's C-terminal region does not affect pilus biogenesis. In addition, our bioinformatic analyses showed that FimT is present across a wide range of γ-Proteobacteria and that the DNA-binding C-terminal region is well-conserved on a sequence level (Fig. 5d). In particular, our alignments of high-confidence FimTs revealed a conserved GRxR motif (where x is often, but not always, a hydrophobic residue) at their C-terminus (Fig. 5d). In

FimT$_{Lp}$ these two arginines correspond to R146 and R148, which we showed contribute to DNA binding in vitro and in vivo (Fig. 4b, c). This motif is less well defined or only partially present in FimU orthologues and those we tested in this study do not bind DNA in vitro (Fig. 5a). Interestingly, a similar C-terminal motif can also be found in the pilins that assemble into the Com pili of Gram-positive organisms, which have been implicated in DNA uptake during natural transformation[26,52,53]. It remains to be investigated, whether this motif also contributes to DNA binding and natural transformation in those proteins.

In summary, this study provides a comprehensive analysis of the molecular mechanisms underpinning FimT's interaction with DNA and demonstrated its pivotal role during natural transformation of the human pathogen *L. pneumophila*. Furthermore, we analysed FimT orthologues from other naturally competent and pathogenic γ-Proteobacteria, which together with our thorough bioinformatic analysis, suggests that FimT is a key player in the natural transformation of a wide range of bacteria.

## Methods

**Bacterial strains and growth conditions.** *L. pneumophila* Lp02 (laboratory strain derived from *L. pneumophila* Philadelphia-1) was cultured in ACES [N-(2-acet-amido)-2-aminoethanesulfonic acid]-buffered yeast extract (AYE) liquid medium or on ACES-buffered charcoal yeast extract (CYE) solid medium, supplemented with 100 μg/mL streptomycin and 100 μg/mL thymidine. When appropriate, chloramphenicol and kanamycin were added at 5 μg/mL and 15 μg/mL, respectively. For the construction of knockout Lp02 strains, the relevant genes and 1000 bp of upstream and downstream regions were first cloned into the pSR47S suicide plasmid (derivative of pSR47[54]). Following deletion of the gene of interest from the plasmid, the modification was introduced onto the Lp02 chromosome by triparental conjugation and subsequent selection as described previously[55,56]. All strains were verified by colony PCR and DNA sequencing (Microsynth) and are listed in Supplementary Table 2.

**Plasmids.** All protein expression constructs were generated using the pOPINS or pOPINB vectors[57,58] carrying an N-terminal His$_6$-SUMO or His$_6$ tag, respectively. Constructs for in vivo studies were generated using pMMB207C[59], by cloning the relevant genes downstream of the P*tac* promoter. DNA fragments were amplified from *L. pneumophila* (RefSeq NC_002942.5) genomic DNA by PCR using CloneAmp HiFi PCR premix (Takara) and the relevant primers. For FimT and FimU orthologues from *P. aeruginosa* PAO1 (RefSeq NC_002516.2) and *X. campestris* pv. *campestris* str. ATCC 33913 (RefSeq NC_003902.1), template DNA was first synthesised (Twist Bioscience). In-Fusion cloning and site-directed mutagenesis was carried out according to the manufacturer's guidelines (Takara). All plasmids and primers used in this study can be found in Supplementary Table 3 and Supplementary Table 4, respectively. A summary of the gene locus tags of genes mentioned in this study from their respective genomes can be found in Supplementary Tables 5 and 6.

**Protein production.** Recombinant His$_6$-SUMO tagged proteins (FimT$_{Lp}$, FimU$_{Lp}$, FimT$_{Pa}$, FimU$_{Pa}$, FimT$_{Xc}$, FimU$_{Xc}$) and His$_6$-tagged proteins (PilA1, PilA2) were expressed in BL21 (DE3) or Shuffle T7 *E. coli* cells (NEB). All constructs were N-terminally truncated to remove the transmembrane helix (α1 N) (Supplementary Table 3). Cultures were grown in Luria-Bertani (LB) media to an optical density at 600 nm (OD$_{600}$) of 0.6–0.8, induced using 0.5 mM IPTG and further incubated at 16 °C for 12–18 h while shaking. Cells were lysed in 50 mM Tris-HCl pH 7.2, 1 M NaCl, 20 mM imidazole, 0.1 mg/mL lysozyme, 1 mg/mL DNAse, and one complete mini EDTA-free protease inhibitor cocktail tablet (Roche), by passing the sample three times through a pressurised cell disruptor (M110-L, Microfluidics) at 12000 psi. The clarified lysate was applied to a 5 mL HisTrap HP column (Cytiva) and His$_6$-SUMO or His$_6$ tagged pilins were eluted with a linear 20–500 mM imidazole gradient. The His$_6$-SUMO or His$_6$ tag was cleaved using the catalytic domain of the human SENP1 protease or PreScission protease, respectively, while the sample was dialysed against 50 mM Tris-HCl pH 7.2, 50 mM NaCl. Protein samples were further purified by cation exchange chromatography using a 5 mL HiTrap SP HP column (Cytiva), from which pilins were eluted using a linear 50–1000 mM NaCl gradient. Lastly, the pilin samples were purified by size exclusion chromatography in 50 mM Tris-HCl pH 7.2, 50 mM NaCl using a HiLoad 16/600 Superdex 75 pg column (Cytiva). Protein samples were concentrated using Amicon Ultra-15 centrifugal filters (3 kDa molecular weight cut-off, Millipore). Reducing agent (2 mM DTT) was included in the buffers for those pilins with free cysteines. All purification steps were performed at 4 °C.

### NMR spectroscopy

*Production of isotope-labelled FimT$_{Lp}$.* To produce uniformly labelled FimT$_{Lp}$, cells were grown in M9 minimal medium containing 1 g/L $^{15}$NH$_4$Cl and further supplemented with 3 g/L glucose (or $^{13}$C$_6$-glucose for double labelled FimT$_{Lp}$), 2 mM MgSO$_4$, trace elements, vitamin mix and appropriate antibiotics for selection. Protein expression was induced at an OD$_{600}$ of 0.6-0.8 with 0.5 mM IPTG and cells were harvested after 20 h at 16 °C. FimT$_{Lp}$ was purified as described above.

*Data acquisition and structure determination.* For resonance assignments and structure determination the following spectra were recorded with a 580 μM sample of uniformly $^{13}$C,$^{15}$N-labelled FimT 28–152 in 25 mM NaP$_i$ pH 7.2, 150 mM NaCl, and 10% D$_2$O at 298 K in a 3 mm diameter NMR tube: 3D HNCACB and 3D CBCACONH spectra[60] were recorded on a 700 MHz AVIIIHD spectrometer equipped with a TCI cryoprobe (Bruker) using the software package Topspin (versions 3.6.2 and 4.0.7). The spectra consisted of 2048 × 50 × 100 complex points in the $^1$H, $^{15}$N and $^{13}$C dimensions with respective spectral widths of 16, 34, and 64 ppm, and were recorded with 8 scans per increment resulting in 2 and 1.5 days of measurement time, respectively. A 3D HcC(aliaro)H-TOCSY[61] was recorded on a 600 MHz AVIIIHD spectrometer equipped with a TCI cryoprobe (Bruker). The spectrum consisted of 1536 × 100 × 150 complex points in the $^1$H, $^1$H, and $^{13}$C dimensions with respective spectral widths of 16, 12, and 140 ppm and was recorded with 2 scans per increment in 3 days using a recycle delay of 2 s. A time shared 3D [$^{13}$C/$^{15}$N,$^1$H]-HSQC NOESY (modified from[62]) was recorded on a 900 MHz AVIIIHD spectrometer equipped with a TCI cryoprobe (Bruker). The spectrum consisted of 1536 × 100 × 256 complex points in the $^1$H, $^1$H, and $^{13}$C/$^{15}$N dimensions with respective spectral widths of 16, 12, and 140/58 ppm and was recorded with 2 scans per increment in 3 days.

Resonance assignments were determined with the program CARA release 1.9.1.8a2 (www.cara.nmr.ch, Keller R (2005), ETH Zürich) to 98.2% completeness. Signals in the NOESY spectra were subsequently automatically picked in the program analysis of the CcpNmr v2.5.1 software package[63]. Peaklists and assignments were used as input for a structure calculation with CYANA v3.98.13[64] where angle constraints were automatically generated from Cα chemical shifts. Manual inspection of the automatically picked peak lists resulted in a set of 4595 picked NOE peaks of which 4220 were assigned in the final CYANA calculation which yielded an average target function value of 0.21. The structures were finally energy minimised in the program Amber20[65]. Statistics for the resulting bundle of 20 conformers can be found in Supplementary Table 1. Additional analysis of the structural bundle after the CYANA calculation revealed 42 hydrogen bonds (each present in more than 6 structures) and the following Ramachandran statistics: 72.2%, 27.4%, and 0.4% of residues in favoured, allowed, and additionally allowed regions, respectively. All structural figures were generated using PyMOL v2.4.1 (https://www.pymol.org).

*DNA binding studies by NMR.* To map the surface patch of FimT$_{Lp}$ involved in DNA binding, chemical shift perturbation experiments were performed using 12 bp or 36 bp dsDNA fragments (Supplementary Table 4). [$^{15}$N,$^1$H]-HSQC experiments of 80 μM $^{15}$N-labelled FimT$_{Lp}$ at saturating concentrations of DNA were recorded. In order to use the same conditions as other assays, all protein and DNA samples for NMR binding studies were dialysed into 50 mM Tris-HCl pH 7.2, 50 mM NaCl buffer. Weighted chemical shift perturbations (CSPs), defined as $((\Delta^1H^2)^{0.5} + ((\Delta^{15}N/5)^2)^{0.5}$ (ppm), were measured by comparing spectra of unbound and bound states. The standard deviation (σ) of the chemical shift range was calculated, CSP maps were plotted in GraphPad Prism v9 and residues for which the shift change was greater than σ were mapped onto the FimT$_{Lp}$ surface. To estimate the equilibrium dissociation constant ($K_D$) of this interaction, [$^{15}$N,$^1$H]-HSQC experiments of 40 μM $^{15}$N-labelled FimT$_{Lp}$ at different concentrations (0–600 μM) of DNA were recorded. For selected residues undergoing large CSPs, binding curves were plotted and fitted to a model assuming one set of binding sites using the software fitKD v1.3.0 (four representative curves are shown in (Supplementary Fig. 5). The spectra were recorded on a 700 MHz AV-NEO spectrometer equipped with a TCI cryoprobe (Bruker) and consisted of 2048×128 complex points using 32 scans per increment resulting in an experiment time of 2 h.

**Electrophoretic mobility shift assay.** Various DNA probes were tested for interaction with purified pilin samples using an agarose gel-based electrophoretic mobility shift assay (EMSA). Short 30 bp dsDNA fragments were generated by annealing complementary 30-meric ssDNA oligonucleotides. All oligonucleotides were obtained from Microsynth and are listed in Supplementary Table 4. The pTRC99A-*lpg2953-2958::Kan* (9074 bp) plasmid, left intact or linearised by a single-cutter restriction enzyme (ClaI), was used for the comparison between circular and linear dsDNA probes, respectively. All DNA probes were resuspended in or dialysed into the same buffer as the protein samples prior to the assay. DNA samples (1 μM of 30-meric ssDNA and dsDNA; 20 ng/μl for longer DNA fragments) were incubated with increasing concentrations (0-100 μM) of pilins in 50 mM Tris-HCl pH 7.2, 50 mM NaCl, 15% (v/v) glycerol in a final volume of 20 μL. These samples were incubated at 25 °C for 30 min and subsequently separated by gel electrophoresis at 10 V/cm for 30 min using 0.9–2.5% (w/v) agarose gels containing SYBR Safe DNA stain (Invitrogen). DNA was visualised using UV illumination in a gel imaging system (Carestream).

**Isothermal titration calorimetry.** Isothermal titration calorimetry (ITC) experiments were carried out in duplicate on a VP-ITC microcalorimeter (MicroCal). All

measurements were performed in 50 mM Tris-HCl pH 7.2, 50 mM NaCl buffer at 30 °C. Following a pre-injection of 1 μL, titrations consisted of 19 consecutive 15 μL injections of 320 μM 12meric dsDNA or 350 μM ssDNA (syringe) into 30 μM FimT$_{Lp}$ (cell) performed at 180 s or 240 s intervals, respectively. The heat of ligand dilution, obtained by injecting DNA into buffer, was subtracted from the reaction heat, and curve fitting was performed in Origin v.7.0383 (OriginLab) using a model assuming two binding sites of equal affinity or "one set" of binding sites.

**Microscale thermophoresis/temperature-related intensity change measurements**. Microscale thermophoresis (MST) experiments were conducted measuring the temperature-related intensity change (TRIC) of the fluorescence signal[66]. A 12 bp fluorescently labelled dsDNA probe was generated by annealing a 5′ fluorescein (FAM)-labelled and an unlabelled strand (Microsynth; Supplementary Table 4) and used in all MST/TRIC experiments. Equilibrium binding assays were performed in 50 mM Tris-HCl pH 7.2, 50–150 mM NaCl, 0.05% (v/v) Tween-20. Increasing concentrations of purified wild-type or mutant FimT$_{Lp}$ samples were incubated with 100 nM of FAM-labelled 12 bp dsDNA probe at 25 °C for 30 min prior to measurement. MST/TRIC measurements were performed at 20 °C using a Monolith NT.115 instrument (NanoTemper) at 25% LED power and 20% MST laser power. Curve fitting was performed with data derived from the TRIC effect using Affinity Analysis (MST) v2.0.2. For the experiment conducted with wild-type FimT$_{Lp}$ measured at 50 mM NaCl, the data appeared slightly biphasic in nature. This suggested the presence of two binding sites with similar, yet non-identical binding affinities. When these data were fitted with a binding model assuming two non-identical binding sites, $K_D$(1) was indeed very similar to that obtained when fit according to two identical sites (~2.9 vs 6.3 μM). All other binding experiments using other methods (ITC and NMR), as well as MST/TRIC experiments conducted with FimT$_{Lp}$ mutants, did not reveal an obvious biphasic binding signature, which could be explained by insufficient resolution. Therefore, we chose to fit all data in the same manner, assuming two identical binding sites, to allow for their comparison. All MST/TRIC measurements were performed at least three times. In addition, all samples were measured twice, 30 min apart, resulting in very similar binding curves and derived dissociation constants, indicating that the binding equilibrium had been attained at the time of measurement.

**Transformation assay**. All transformation assays were performed with the *L. pneumophila* Lp02 strain in liquid medium at 30 °C, similar to transformation assays performed by others[10,67]. Strains were streaked onto CYE solid medium from frozen stocks and incubated at 37 °C for 3-4 days. From this plate, bacteria were resuspended in a liquid starter culture (5 mL of AYE medium) and incubated at 37 °C overnight while shaking at 200 rpm. The starter culture was diluted into a fresh 10 mL AYE culture (starting OD$_{600}$ of 0.02) and cultured at 30 °C while shaking. Once the culture reached an OD$_{600}$ of 0.3, 1 mL was transferred into a new tube and incubated with 1 μg of transforming DNA at 30 °C for a further 24 h. The transforming DNA consisted of a 4906 bp PCR product encompassing the *L. pneumophila* genomic region spanning *lpg2953-2958*, where the *hipB* gene (*lpg2955*) is interrupted by a kanamycin resistance cassette (based on[68]). This provides 2000 bp regions of homology up- and downstream of the resistance cassette. Tenfold serial dilutions of the culture were plated on selective (supplemented with 15 μg/mL kanamycin) and non-selective CYE media. The plates were incubated at 37 °C for 4–5 days and colony forming units (CFUs) were counted. The transformation efficiency corresponds to the ratio of the number of CFUs obtained on selective medium divided by the number of CFUs counted on non-selective medium. The minimum counting threshold was set at 10 colonies per plate. Transformation assays to test complementation of knockout strains with protein ectopically expressed from the pMMB207C plasmid were performed in the same manner, except for the addition of 0.5 mM IPTG during the incubation step of the bacteria with transforming DNA. Transformation assays requiring direct comparison between strains or complemented strains were performed in parallel.

**Bioinformatic analyses**

*Collection of putative FimT and FimU sequences*. Three sets of FimT or FimU sequences were collected as follows: (1) a FimT set was retrieved by BlastP (Blast v2.11.0) against FimT$_{Lp}$, FimT$_{Pa}$, FimT$_{Ab}$ and FimT$_{Xc}$ with a 95% query coverage cutoff, (2) a FimU set was retrieved by BlastP against FimU$_{Lp}$, FimU$_{Pa}$, FimU$_{Ab}$ and FimU$_{Xc}$ with a 95% (Pa, Ab, Xc) or 80% (Lp) query coverage cutoff, (3) a diverse FimT/U set was retrieved by a PSI-blast[69] search against FimT$_{Lp}$, with >95% query coverage and e-value >0.005 cutoffs applied at each iteration, and the search continued for 8 iterations. To limit redundancy in the results all searches were conducted against the refseq_select protein database which, for prokaryotes, contains only sequences from representative and reference genomes. The FimT and FimU sets were used for initial gene neighbourhood analyses beyond the four functionally characterised representatives (Supplementary Fig. 6), while the diverse set was used for phylogenetic analysis and to define conserved regions.

*Gene neighbourhood analysis*. The gene neighbourhood of each *fimT* and *fimU* was examined using custom Biopython[70] scripts and NCBI resources as follows 1) source genome(s) for each protein entry were identified from the Identical Protein Groups (IPG) resource (this was necessary because many of the blast results were non-redundant entries comprising multiple identical proteins), 2) the genome region

corresponding to the gene of interest and 5000 bp up- and downstream was downloaded from the nucleotide database for one representative of each IPG (if < 5000 bp flanking up- and downstream sequence was available the entry was excluded from further analysis), and 3) coding sequences in the neighbouring region were extracted as multifasta and searched against the Pfam[71] database of domain profiles using HMMER v3.3.1[72] (hmm scan function, e-value threshold 0.0001). *fimT* or *fimU* genes were classified as orphans or minor pilin locus components based on the presence of one or more of the Pfam domains PilC, PilX, PilX_N and PilW in the flanking region. The presence of just one of these domains was defined as indicating a minor pilin locus, to account for the possibility that proteins only weakly matching the relevant Pfam domain would be missed, or that relevant proteins may be found >5000 bp away. NCBI scripts used in this study are available at Zenodo (https://doi.org/10.5281/zenodo.5920973)[73].

*Generation of high-confidence FimT set and phylogenetic analysis*. Because FimT is a GspH-domain protein and shares overall structural similarity with the type IV minor pilin FimU and the T2SS protein GspH, putative homologues from the diverse FimT/U set were filtered based on their gene neighbourhood to exclude likely *fimU* genes and generate a subset of high-confidence putative *fimT* genes for further analyses. As 100% of genes in the FimU set were located in minor pilin operons, orphan genes within the diverse FimT/U set were presumed to encode genuine FimT proteins, and these sequences were aligned along with FimT$_{Lp}$, FimT$_{Pa}$, FimT$_{Ab}$ and FimT$_{Xc}$ and used to generate a FimT HMM profile using HMMER[72] (hmmbuild function). A FimU HMM profile was generated from sequence set 2 (FimU homologues), following alignment with MUSCLE v3.8.31[74] and removal of entries showing >80% amino acid identity to another entry. Each sequence from the diverse FimT/U set was scanned against both the FimU and FimT sequence HMMs and reported as a likely FimT if its match score to the FimT profile was >20 points greater than its match to the FimU profile. In this way, a set of 196 putative FimT protein sequences was obtained. FimT protein sequences were aligned using MUSCLE with default (high-accuracy) settings, and the alignment was visualised and manually improved using JalView v2.11.1.7[75]. The FimT alignment was processed using TrimAL v1.4.1[76] to remove low-quality positions and uninformative sequences (parameters: -strictplus -resoverlap 0.8 -seqoverlap 75). A maximum-likelihood phylogenetic tree of the FimT homologues was constructed using IQtree v1.6.12[77] with the substitution model LG + F + R5[78] and ultrafast bootstrapping[79]. The phylogenetic tree and associated metadata was viewed using iTol v5[80]. The tree was midpoint-rooted and branches with less than 50% bootstrap support removed. Gene neighbourhood diagrams for selected FimT homologues were generated using Clinker v0.0.20[81]. The FimT motif diagram was generated using WebLogo v2.8.2[82].

**Reporting summary**. Further information on research design is available in the Nature Research Reporting Summary linked to this article.

## Data availability
NMR spectra and corresponding model coordinates have been deposited in the BioMag Resonance Data Bank (BMRB: 34704) and Protein Data Bank (PDB ID: 7QYI), respectively. Any other data that support the findings of this study are available from the corresponding author upon request. Source data are provided with this paper.

## Code availability
The code used in this study has been deposited in the GitHub repository (https://github.com/francesca-short/NCBI_scripts)[73].

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

## Acknowledgements

This work was funded by an SNSF PRIMA grant PR00P3_179728 to M.K.H. F.L.S. is supported by an Australian Research Council Discovery Early Career Research Award DE200101524. We would like to thank G. Waksman and A. Meir for the Lp02, CR019 and DH5α λpir strains, and the pSR47S plasmid. We would also like to thank H. Hilbi for the pMMB207C plasmid. We are grateful to R. Glockshuber, E. Weber-Ban, and M. Pilhofer for helpful discussions and sharing of reagents and instruments, and to J. Scheuermann for the use of the VP-ITC instrument.

## Author contributions

SAGB cloned constructs, created *Legionella* strains, purified proteins, performed DNA binding studies, transformation assays, Western blots, and analysed results. FLS designed and performed all bioinformatic analyses. SH constructed FimT chimera constructs and performed the corresponding transformation assays. MJMS purified proteins and performed ITC experiments. ADG performed and analysed all NMR-related experiments with help from SAGB. MKH designed and supervised the study, made figures and wrote the manuscript with help from all authors.

## Competing interests

The authors declare no competing interests.
