## [Peer Review File · Nature Communications]

Reviewers' Comments:

Reviewer #1:

Remarks to the Author:

This manuscript by Braus et al reports the ability of a type IV pilin subunit, FimT, from *Legionella pneumophila* to bind to DNA in vitro, through multiple quantitative methods, as well as molecular evidence that deletion or specific mutation in FimT significantly reduces natural competence in *L. pneumophila*. Additionally, the authors have resolved the three-dimensional structure of FimT in solution by NMR and are able to directly observe residue-specific shifts upon titration of DNA to infer the site of DNA-binding to be in the C-terminus (rather than the more central interface found in ComP). Based on the homology between *L. pneumophila* FimT and pilins from other gram-negative bacterial species, they investigated the propensity for DNA-binding by FimT orthologues from *P. aeruginosa* and *X. campestris* and found that they also bound dsDNA robustly in vitro. Taken together, these results imply that DNA-recognition by FimT-like proteins may be general to many phyla of bacteria and represent an important, possibly essential, mechanism for DNA uptake for natural competence for those species.

These data are exciting and represent a substantial advance for the field; previously only single natural competence receptor has been definitively identified and did not allow for the identification of similar DNA-binding receptors in other taxa. Here, we are presented with not only the identification of a DNA-receptor which may well be general to multiple species but structural data which provide some insight into the mechanism of DNA-recognition. I believe this manuscript is well-written and conclusions well-supported by the data. I have only minor comments.

Specific comments:

- For readers unfamiliar with the *L. pneumophila* T4P system, it would be helpful to have some sort of list in the text or diagram in a figure showing all the putative subunits. I infer from the combination of Figure 1 and extended Figure 6 that there are eight (PilA1, PilA2, FimU, FimT, PilV, PilW, PilX and PilE), four of which were used in the initial EMSA assays.
- As the authors point out in the text, it would appear that some binding is observed between the DNA-sample and the *X. campestris* FimU (Figure 5), albeit at a lower affinity than any of the FimT samples (it is difficult to compare them more precisely with these concentration points). Does FimUXC have a C-terminus which is more basic (or FimT-like in other respects) than the other two FimU proteins in that experiment?
- Because both pilus assembly and pilus retraction are required for DNA-uptake in many bacterial species, any evidence that the fimT mutant is capable of pilus retraction would strengthen the conclusion that the knockout reduces competence solely through a reduction in DNA binding.
- I could not find any comment in the text, but in the transverse relaxation data in Extended Data Figure 3, the C-terminal residue (or possibly the final two residues) appears to have a much higher T2 value than the average; reversing the trend the authors mention for residues 140-150. Is this a general feature of disordered C-termini or to be expected based on other factors?

Reviewer #2:

Remarks to the Author:

This is a well-presented manuscript; the experiments have been carefully designed and executed to a high standard, although there are some limitations to the study. The identity of the primary DNA receptor(s) for natural competence/transformation has been an important topic in this field for several years. Several lines of evidence point to minor pilins in type IV pilus biogenesis having specific functional roles, in adhesion or DNA binding for example. The current model for DNA uptake requires recognition of nucleic acid outside the outer membrane and therefore needs at least one receptor which is associated with the pilus in some way. Although some components of the type IV pilus biogenesis system are well conserved, there does not appear to be a unique pilin-like protein which satisfies these criteria in all Gram negative bacteria. As the authors state, the

most advanced work to date on this question has been carried out on the ComP system in *Neisseria*, but there do not appear to be any ComP-like counterparts in other bacteria. The importance of the manuscript therefore lies in the proposal that FimT is a DNA-specific receptor in *Legionella* which functions in this way in other gamma-proteobacteria. The evidence that FimT is a DNA receptor in *Legionella* is sound and well presented. Based on these observations, and conservation of sequence and structure, it is reasonable to propose that FimT orthologs in other species might function in a similar way. However, the manuscript falls short by not formally demonstrating this in *Pseudomonas*, *Xylella* and other gamma-proteobacteria.

1. Is there evidence that FimT is incorporated into the assembled pilus fiber (eg from a Western blot of isolated pili using anti-FimT antibody)?
2. A major point, which is not well covered in the Results or Discussion sections, is that FimT would presumably function within an assembled pilus, rather than as a monomer. This has important ramifications. For example, the NMR data establishes that the C-terminus (140-152) is more dynamic and less structured. But how would this region of FimT behave in an assembled pilus? Some modelling of pilus assembly might help to establish if it is more structured in the assembled state (there is a model of *Pseudomonas* type IV pili PDB 5vxy)
3. The same limitation applies to the DNA binding experiments. In the section on interface mutations (L223 onwards), we would expect extracted pili from these mutants to show reduced DNA binding- is that the case?
4. Have the authors attempted to model the FimT:DNA complex, both as a monomer and in a model of the pilus fiber?
5. The identification of a conserved Arg motif in the FimT C-terminal region is interesting (Fig 5d); however it is possible that this region is structured differently in orthologs with the disulphide bond (L345 'it is likely that disulphide bond-containing FimT orthologues do not possess a conformationally flexible C-terminal tail'). Would this alter the way in which DNA interacts with the C-terminal region in these variants?
6. In the discussion, it would be useful to distinguish between DNA receptors which are proposed to lie at the tip of the pilus (eg ComZ) and other pilin-like proteins which are incorporated into the main body of the pilus fiber (eg ComP).

Reviewer #3:

Remarks to the Author:

The study by Braus et al. identifies that FimT is a DNA-binding minor pilin that contributes to DNA uptake during natural transformation in *Legionella pneumophila*. They solve the solution structure of FimT and analyze the potential residues that may be involved in DNA-binding. They go on to test the relative impact of these residues on DNA binding and natural transformation both biochemically using purified FimT and in vivo via transformation assays. Finally, they demonstrate that FimT orthologs may also play a role in DNA binding in other systems.

The molecular mechanism underlying how competence T4P bind to DNA has remained very poorly characterized – especially in the vast majority of systems that bind to DNA in a sequence-independent manner. This study provides a major advance in our understanding of this process and should be broadly interesting to the field. The experiments are conducted rigorously for the most part and the manuscript is well-written and organized. I have only two major points for the authors to consider.

Major:

Fig. S1A – The observation that FimT and FimU are both dispensable for pilus assembly would be quite interesting. But it is unclear how these assays were controlled to demonstrate that the PilA2-FLAG signal in the supernatant samples is attributed to sheared surface pili. To demonstrate that

the signal in the supernatants is not simply due to cell lysis, blots could be assessed for a cytoplasmic marker. Or perhaps better would be to perform this analysis on a mutant strain that cannot assemble pili, like a pilB or pilQ deletion. If these assays truly report on sheared surface pili, this control strain should have no PilA2-FLAG signal in the supernatant fractions.

Fig. 4b-c – it is interesting that the point mutations diminish DNA binding of FimT only 3-10 fold, while the effect on natural transformation in vivo is greater than 100-fold. Since FimT is potentially dispensable for pilus assembly, isn't it feasible that that these point mutations prevent FimT from being assembled into the pilus filament? There is no simple way to directly assess minor pilin incorporation into the pilus filament that I am aware of because as little as a single copy of each minor pilin may be within the filament, which makes detecting these minor pilins difficult via Western blotting or similar approaches. So at the very least, the authors should acknowledge this potential alternative explanation in the text. Alternatively, the authors may want to consider testing whether FimT and FimU are genetically redundant for pilus assembly as previously hypothesized (in ref 35). If a fimU fimT double mutant does not assemble pili, then it is possible that pilus assembly can be complemented by just fimT. By assessing pilus assembly in a fimU mutant, this may provide an experimental basis to test whether FimT point mutants can support pilus assembly, and therefore, be incorporated into the pilus filament.

Fig. 5b – absence of complementation with FimT orthologs from Pa and Xc could result either because these proteins do not assemble into the legionella T4P, or because they do not natively promote DNA-binding (although the chimeric construct does demonstrate that the C-terminal portion of the Pa FimT is capable of promoting DNA binding when appended to the end of Lp FimT). Again, this should be acknowledged in the text or tested further as discussed above.

Minor:

Line 129 – "...complete loss of natural transformation..."

Line 329-330 – the authors should elaborate on why they believe this 2:1 interaction is not likely to be physiologically relevant. Because only a single copy of FimT is hypothesized to be at the pilus tip? Or conformationally even if there were two copies of FimT in the pilus, they would not be able to bind both sides of dsDNA?

Line 359-360 – change "...on the other hand, both proteins showed near wild-type levels of twitching." to "...on the other hand, loss of either protein showed near wild-type levels of twitching."

REVIEWER COMMENTS

Reviewer #1 (Remarks to the Author):

This manuscript by Braus et al reports the ability of a type IV pilin subunit, FimT, from *Legionella pneumophila* to bind to DNA in vitro, through multiple quantitative methods, as well as molecular evidence that deletion or specific mutation in FimT significantly reduces natural competence in *L. pneumophila*. Additionally, the authors have resolved the three-dimensional structure of FimT in solution by NMR and are able to directly observe residue-specific shifts upon titration of DNA to infer the site of DNA-binding to be in the C-terminus (rather than the more central interface found in ComP). Based on the homology between *L. pneumophila* FimT and pilins from other gram-negative bacterial species, they investigated the propensity for DNA-binding by FimT orthologues from *P. aeruginosa* and *X. campestris* and found that they also bound dsDNA robustly in vitro. Taken together, these results imply that DNA-recognition by FimT-like proteins may be general to many phyla of bacteria and represent an important, possibly essential, mechanism for DNA uptake for natural competence for those species.

These data are exciting and represent a substantial advance for the field; previously only single natural competence receptor has been definitively identified and did not allow for the identification of similar DNA-binding receptors in other taxa. Here, we are presented with not only the identification of a DNA-receptor which may well be general to multiple species but structural data which provide some insight into the mechanism of DNA-recognition. I believe this manuscript is well-written and conclusions well-supported by the data. I have only minor comments.

Specific comments:

1. For readers unfamiliar with the *L. pneumophila* T4P system, it would be helpful to have some sort of list in the text or diagram in a figure showing all the putative subunits. I infer from the combination of Figure 1 and extended Figure 6 that there are eight (PilA1, PilA2, FimU, FimT, PilV, PilW, PilX and PilE), four of which were used in the initial EMSA assays.

We thank the reviewer for this suggestion. We have included a description of the known *Legionella* T4P pilins in the Results section (Lines 109-118).

2. As the authors point out in the text, it would appear that some binding is observed between the DNA-sample and the *X. campestris* FimU (Figure 5), albeit at a lower affinity than any of the FimT samples (it is difficult to compare them more precisely with these concentration points). Does FimUXC have a C-terminus which is more basic (or FimT-like in other respects) than the other two FimU proteins in that experiment?

FimT and FimU are homologues and their deconvolution at the sequence level is not trivial. We set out to identify FimT/FimU in a number of species using a bioinformatic approach that takes into account: i) the genomic neighbourhood, ii) the amino acid sequence, and iii) the available functional information from *L. pneumophila*, *P. aeruginosa*, *X. campestris* and *A. baylyi*. This highlighted a GRxR motif near the C-terminus, which is well-conserved amongst the FimTs and less well defined or only partially present in most FimU orthologues. FimU from *L. pneumophila* and *A. baylyi*, both contain the highly conserved Gly (position 171 and

138, respectively), yet lack both Arg residues. Like several of our high confidence FimUs (Source Data: multisequence alignment), FimU from *P. aeruginosa* contains the Gly (position 153) and the first Arg (position 154). FimU from *X. campestris* contains the Gly (position 161) and the second Arg (position 164) in the degenerated GRxR motif, as well as two other nearby positively charged residues (Arg 166 and Lys 167). This may indeed be the reason that FimU_{Xc} shows weak binding at high protein concentrations. We have added the following sentence to the text: “This may be due to FimU_{Xc}’s slightly more basic C-terminal tail region, and/or other the presence of other residues elsewhere in the protein that allow FimU_{Xc} to weakly interact with DNA.” (Lines 282-284)

Indeed some of the FimU hits contain both Arg residues of the GRxR motif (e.g. *Marinobacter psychrophilus*, *Nitrosomonas ureae*, *Azoarcus pumilus* and others). Yet, in our view, this does not necessarily mean that these FimU orthologues are capable of robust DNA binding. This is because, in the case of FimT, we showed that several other regions (in particular the β 4- β 5, β 5- β 6 loops) are also involved in DNA-binding by forming an overall electropositive surface patch. It remains to be tested whether some of the lower confidence FimT and FimU hits in our list show some degree of cross-over in their DNA binding activity.

3. Because both pilus assembly and pilus retraction are required for DNA-uptake in many bacterial species, any evidence that the fimT mutant is capable of pilus retraction would strengthen the conclusion that the knockout reduces competence solely through a reduction in DNA binding.

We agree with the reviewer’s assessment. However, it is very difficult to directly observe pili (e.g. for live cell fluorescence microscopy) in our *Legionella* system, as our strain (Lp02) does not appear to display many T4P fibres in total. We have looked at many cells by negative stain electron microscopy (NS-EM), and only ever observed a handful of pili, even in the Δ *pilT* background. This is not necessarily surprising, as we believe that low levels of piliation are still consistent with the transformation frequencies we observe. However, this complicates any experiment with the goal of directly observing retraction.

Instead we have added a control experiment (Figure 4d), where we have mutated residues not involved in DNA binding, but within the same three regions we identified to contribute to DNA binding, and show that *in vivo* natural transformation levels are unaffected (in contrast to our mutants that inhibit DNA binding). We believe this experiment shows that the reduced transformation levels of the FimT DNA binding mutants are a direct consequence of their diminished ability to bind to DNA.

4. I could not find any comment in the text, but in the transverse relaxation data in Extended Data Figure 3, the C-terminal residue (or possibly the final two residues) appears to have a much higher T₂ value than the average; reversing the trend the authors mention for residues 140-150. Is this a general feature of disordered C-termini or to be expected based on other factors?

We thank the reviewer for pointing this out, as the contrast between the final two residues and the rest of the C-terminus helps to further clarify the special properties of residues 140–150. For disordered N- and C-terminal tails of proteins one indeed expects very high T₂ values, as flexible tails have similar dynamic properties as small molecules. Here, only the final two residues exhibit such properties, e.g. high T₂ values and negative heteronuclear

NOE values. The conclusion we therefore draw is that only the two final residues are fully flexibly disordered, while the rest of the C-terminus is undergoing dynamical exchange between relatively well-ordered states on a much slower timescale. We added the following sentence to the legend of Extended Data Figure 3: “Only the final residue exhibits a long T₂ values as typical for flexibly disordered residues.”

Reviewer #2 (Remarks to the Author):

This is a well-presented manuscript; the experiments have been carefully designed and executed to a high standard, although there are some limitations to the study. The identity of the primary DNA receptor(s) for natural competence/transformation has been an important topic in this field for several years. Several lines of evidence point to minor pilins in type IV pilus biogenesis having specific functional roles, in adhesion or DNA binding for example. The current model for DNA uptake requires recognition of nucleic acid outside the outer membrane and therefore needs at least one receptor which is associated with the pilus in some way. Although some components of the type IV pilus biogenesis system are well conserved, there does not appear to be a unique pilin-like protein which satisfies these criteria in all Gram negative bacteria. As the authors state, the most advanced work to date on this question has been carried out on the ComP system in *Neisseria*, but there do not appear to be any ComP-like counterparts in other bacteria. The importance of the manuscript therefore lies in the proposal that FimT is a DNA-specific receptor in *Legionella* which functions in this way in other gamma-proteobacteria. The evidence that FimT is a DNA receptor in *Legionella* is sound and well presented. Based on these observations, and conservation of sequence and structure, it is reasonable to propose that FimT orthologs in other species might function in a similar way. However, the manuscript falls short by not formally demonstrating this in *Pseudomonas*, *Xylella* and other gamma-proteobacteria.

We would have liked to expand our study to include experiments in these organisms, but this was not within the scope of the current work, as we would have needed to establish all methods (culturing, genetic modification, transformation assays etc.) for several additional pathogens in our laboratory. Instead we have included the experiments of the FimT chimeras (Fig. 5b), which suggest that FimT orthologues from other bacteria have the same *in vivo* function. In addition, there is evidence that FimT deletion leads to a natural transformation phenotype in *A. baylyi* (Leong et al, 2017).

1. Is there evidence that FimT is incorporated into the assembled pilus fiber (eg from a Western blot of isolated pili using anti-FimT antibody)?

We have attempted this using several approaches. First, we tried to detect C-terminally tagged FimT (Flag or HA-tagged) from fractions that we sheared from the bacterial cell surface via Western blotting. Second, we have now also raised a new antibody against FimT itself (as suggested) and attempted similar Western blotting approaches. Third, we have analysed sheared surface fractions of wild-type cells and cells expressing tagged FimT by mass spectrometry. Lastly, we have designed an enrichment strategy, where we pull down overexpressed Flag-tagged Pila2 from sheared surface fractions before attempting to detect FimT, either by Western blotting or mass spectrometry. For the Western blotting approaches, we have additionally tried to use both native promoter regions, as well as overexpression strategies, where expression was driven by an IPTG-inducible promoter. However, with all of these approaches, we were never able to detect FimT in sheared fractions confidently.

We believe that this is due to two compounding factors. First, our strain of *L. pneumophila* cells (Lp02) does not appear to produce many T4P visible on the cell surface (see our response to Reviewer 1, point 3). Second, it is possible that FimT is only present as a single copy within each T4P fibre (1 subunit out of 1000s). Therefore, this very low abundance makes it extremely challenging to detect FimT within a T4P fibre.

2. A major point, which is not well covered in the Results or Discussion sections, is that FimT would presumably function within an assembled pilus, rather than as a monomer. This has important ramifications. For example, the NMR data establishes that the C-terminus (140-152) is more dynamic and less structured. But how would this region of FimT behave in an assembled pilus? Some modelling of pilus assembly might help to establish if it is more structured in the assembled state (there is a model of *Pseudomonas* type IV pili PDB 5vxy)

We thank the reviewer for this suggestion. The difficulty at this point is that it is not yet clear whether FimT is incorporated throughout the pilus fibre or at the pilus tip, and how many copies of FimT are present in each pilus. Nevertheless, we have generated *in silico* models of FimT in the context of an assembled type IV pilus (within the fibre and at the tip), based on the available structures of the *Pseudomonas* type IV filaments (PDB 5vxy). Modelling FimT at the tip is challenging, given that no structures of the other minor pilins that likely constitute the pilus tip complex are available. For now, we have simply modelled FimT at the very top of the filament. Our models suggest that DNA binding is possible, regardless of whether FimT is incorporated within the fibre or at the tip, as the C-terminus is predicted to face outward even if FimT is located within the body of the pilus fibre. We cannot comment at this point whether or not the conformationally flexible C-terminal region would adopt a more structured conformation once FimT is incorporated into a pilus. For these reasons we feel that the *in silico* models we have generated are too speculative to merit inclusion in the manuscript, but we show these models in Figure 1 below for inspection:

Figure 1: Packing of FimT_{Lp} and FimT_{Lp}-DNA into a T4P

a-d, Models of FimT_{Lp} and the FimT_{Lp}-DNA complex generated by Haddock packed into the assembled PAK pilus (PDB ID: 5VXY) shown in two orientations rotated by 120°. A pilus subunit (PilA) of the PAK pilus was replaced by the structural ensemble of the 20 lowest energy NMR structures of FimT_{Lp} (a, b) or the FimT_{Lp}-DNA complex (c, d), either within the pilus fibre (a, c), or at the pilus tip (b, d).

3. The same limitation applies to the DNA binding experiments. In the section on interface mutations (L223 onwards), we would expect extracted pili from these mutants to show reduced DNA binding- is that the case?

This is a very good point and we agree with the reviewer's conclusion. We have made several attempts to purify enough pili for such an experiment. However, such an experiment is extremely challenging for the same reasons as explained for point 1 above, namely the very low number of pili displayed on the *L. pneumophila* Lp02 cell surface. We believe that our *in vitro* DNA binding experiments are extremely comprehensive and would therefore predict that extracted pili recapitulate what we see for FimT on its own.

4. Have the authors attempted to model the FimT:DNA complex, both as a monomer and in a model of the pilus fiber?

We thank the reviewer for this suggestion. Using our NMR chemical shift perturbation findings about which residues are involved in DNA binding, we have generated a model of the FimT_{Lp}:DNA complex, as well as complexes of three FimT orthologues bound to DNA, using Haddock (see point 3 below, Figure 2). We positioned and modelled the FimT_{Lp}:DNA complex in the context of a pilus using PDB 5vxy, located both within the pilus fibre or at the tip (see point 2 above).

5. The identification of a conserved Arg motif in the FimT C-terminal region is interesting (Fig 5d); however it is possible that this region is structured differently in orthologs with the disulphide bond (L345 'it is likely that disulphide bond-containing FimT orthologues do not possess a conformationally flexible C-terminal tail'). Would this alter the way in which DNA interacts with the C-terminal region in these variants?

Figure 2: *In silico* models of FimT_{Lp} and FimT orthologues in complex with DNA
a, *In silico* Haddock model of FimT_{Lp} and the AlphaFold structural models of FimT_{Pa} (**b**), FimT_{Xc} (**c**) and FimT_{Ab} (**d**) in complex with a 12meric dsDNA molecule.

In addition to modelling the complex of DNA with the *L. pneumophila* FimT orthologue (Figure 2a), we also modelled the *P. aeruginosa*, *X. campestris* and *A. baylyi* FimT orthologues (Figure 2b-d), which all contain the D-region defining disulphide bond, bound to DNA. To do this, we first generated structural predictions of these orthologues using AlphaFold, and predicted potential residues involved in DNA binding for the Haddock modelling using our knowledge of how the *Legionella* version of FimT binds to DNA together with information on the conservation of these residues. Although these models look

plausible and could show how both a conformationally flexible, as well as a more structured, C-terminal region is involved in DNA binding, we would like to stress that caution needs to be applied to the interpretation of these models. We find it striking that the angle at which the DNA molecule contacts FimT is extremely different between the models. We believe that future experimental structures of FimT-DNA will be essential to reveal how FimT precisely binds to DNA. In addition, further knowledge about FimT's localisation and copy number within the assembled pilus fibre (tip or elsewhere) will be required to fully understand this process in the context of an assembled pilus. For now, we fear that such models are too speculative for inclusion in the manuscript.

We have added the following sentence to the discussion “It remains to be seen whether or not the C-terminal tail of FimT_{LP} adopts a more structured conformation in complex with DNA and/or in the context of a fully assembled pilus.” (Lines 387-389)

6. In the discussion, it would be useful to distinguish between DNA receptors which are proposed to lie at the tip of the pilus (eg ComZ) and other pilin-like proteins which are incorporated into the main body of the pilus fiber (eg ComP).

We have clarified our discussion about DNA receptors located at the pilus tip vs within the pilus fibre in the text (Lines 433-439).

Reviewer #3 (Remarks to the Author):

The study by Braus et al. identifies that FimT is a DNA-binding minor pilin that contributes to DNA uptake during natural transformation in *Legionella pneumophila*. They solve the solution structure of FimT and analyze the potential residues that may be involved in DNA-binding. They go on to test the relative impact of these residues on DNA binding and natural transformation both biochemically using purified FimT and in vivo via transformation assays. Finally, they demonstrate that FimT orthologs may also play a role in DNA binding in other systems.

The molecular mechanism underlying how competence T4P bind to DNA has remained very poorly characterized – especially in the vast majority of systems that bind to DNA in a sequence-independent manner. This study provides a major advance in our understanding of this process and should be broadly interesting to the field. The experiments are conducted rigorously for the most part and the manuscript is well-written and organized. I have only two major points for the authors to consider.

Major:

1. Fig. S1A – The observation that FimT and FimU are both dispensable for pilus assembly would be quite interesting. But it is unclear how these assays were controlled to demonstrate that the PilA2-FLAG signal in the supernatant samples is attributed to sheared surface pili. To demonstrate that the signal in the supernatants is not simply due to cell lysis, blots could be assessed for a cytoplasmic marker. Or perhaps better would be to perform this analysis on a mutant strain that cannot assemble pili, like a pilB or pilQ deletion. If these assays truly report on sheared surface pili, this control strain should have no PilA2-FLAG signal in the supernatant fractions.

We thank the reviewer for pointing out this essential control. We have repeated these experiments using *pilQ* and *pilT* deletion strains and a cytoplasmic marker as controls and indeed observed some degree of lysis when PilA2-FLAG was overexpressed from a plasmid. This was surprising to us, given that a the same experiment conducted in another *Legionella* strain (published elsewhere) did not show this problem. Nevertheless, we have now removed this Western blot. In an attempt to repeat this experiment, we then generated a strain that encodes FLAG-tagged PilA2 directly in the genome of *L. pneumophila* (subject to native regulation) as we thought that PilA2 overexpression is causing the lysis issue we observed. Indeed, we no longer observe lysis in this set up, yet unfortunately struggle to detect PilA2-FLAG in the sheared fractions. On one occasion we have detected a similar amount of signal for wild-type, $\Delta fimT$, $\Delta fimU$, $\Delta fimT/fimU$, no signal in the $\Delta pilQ$ control, and a slight increase for the $\Delta pilT$ control (as expected). Yet on most occasions we have not detected any signal except for in the *pilT* deletion background. We have also looked at Lp02 cells (wild-type and $\Delta pilT$) overexpressing PilA2-Flag by negative stain electron microscopy and observed very few T4P, further explaining the difficulties with this Western blot. Although our new data provide a *hint* that pilus biogenesis still occurs in $\Delta fimT$, $\Delta fimU$, $\Delta fimT/fimU$ strains, they are not yet conclusive and we prefer to remain cautious with our interpretation.

We have added the sentence “Although the absence of FimT does not appear to affect pilus biogenesis or other T4P functions such as twitching motility in other organisms, this possibility cannot be ruled out.” (Lines 140-142) after our discussion of the *in vivo* transformation assay results presented in Figure 1. We have also included additional controls and expanded our discussion for the follow up experiment with the FimT DNA binding mutants (Figure 4). Please see our answer to the next point (2) below.

2. Fig. 4b-c – it is interesting that the point mutations diminish DNA binding of FimT only 3-10 fold, while the effect on natural transformation *in vivo* is greater than 100-fold. Since FimT is potentially dispensable for pilus assembly, isn't it feasible that that these point mutations prevent FimT from being assembled into the pilus filament? There is no simple way to directly assess minor pilin incorporation into the pilus filament that I am aware of because as little as a single copy of each minor pilin may be within the filament, which makes detecting these minor pilins difficult via Western blotting or similar approaches. So at the very least, the authors should acknowledge this potential alternative explanation in the text.

The reviewer raises an interesting point. We initially embarked on the *in vivo* experiments with point mutations in FimT, specifically because we believe it to be highly unlikely that a single point mutation would prevent pilus assembly. To further strengthen this point, we have conducted an additional control experiment (now Figure 4d) where we have mutated residues not involved in DNA binding, but within the same three regions ($\beta 4$ - $\beta 5$ loop, $\beta 5$ - $\beta 6$ loop and the C-terminal tail) previously identified to contribute to the DNA binding, and show that natural transformation levels are unaffected. We believe this makes it very unlikely that the phenotype produced by the DNA binding mutants is due to a pilus assembly defect. We have improved our discussion of this in the text (Lines 255-271).

In addition to this, there are several other reasons as to why we think our FimT mutants are properly assembled and incorporated into a pilus:

- Our chimeric FimT construct (exchanged >30 residues at the C-terminus) is functional *in vivo*. This suggests that a more significant perturbation in this region also does not prevent pilus assembly.
- Different FimT mutations do not produce an identical effect, but the *in vivo* effect is greater for the triple and double mutants, compared to the single mutants, meaning there is a rough correlation between *in vitro* affinity and *in vivo* effect.
- The residues we have chosen to mutate are not *absolutely* conserved in all FimT/FimU homologues (see also our response to point 2, Reviewer 1), which would suggest that they are not required for pilus assembly.
- We were able to express and purify mutants for our *in vitro* MST experiments and all samples were monodisperse and highly stable even at high concentrations.
- *In silico* structural models shows that these point mutations are not near any critical inter-subunit interfaces within the assembled pilus filament, and their location in the conformationally flexible C-terminus makes a role in pilus assembly unlikely (also C-terminal globular domains are generally diverse, even within a single organism).

As for a possible reason that could explain the discrepancy between a modest reduction in binding affinity *in vitro* and the more pronounced effect *in vivo*, we believe it is difficult to rationalise this observation because the *in vivo* experiment simply represents a much more complex environment. Nevertheless, plausible explanations could include that there might be an additional avidity effect *in vivo* if each pilus contains >1 copy of FimT, something we currently do not know. And/or, that the association (k_{on}) and dissociation rates (k_{off}) are such that a pilus containing wild-type FimT achieves and maintains DNA binding sufficiently to bring it towards the OM for subsequent uptake steps, even though the pilus extends and retracts in the order of seconds. We currently suspect that the FimT R to Q mutants have a reduced k_{on} rate, which may explain the difficulty in DNA binding to a dynamic pilus containing mutant FimT. Importantly, these factors do not play a role in our *in vitro* DNA binding experiment, which are equilibrium binding MST experiments.

3. Alternatively, the authors may want to consider testing whether FimT and FimU are genetically redundant for pilus assembly as previously hypothesized (in ref 35). If a *fimU fimT* double mutant does not assemble pili, then it is possible that pilus assembly can be complemented by just *fimT*. By assessing pilus assembly in a *fimU* mutant, this may provide an experimental basis to test whether FimT point mutants can support pilus assembly, and therefore, be incorporated into the pilus filament.

We very much agree with Reviewer 3's assessment and believe that a lot remains to be learned about the interplay between FimU and FimT, with regards to their genetic regulation and assembly into T4P filaments. Observing FimT and FimU directly in pili is challenging in *Legionella* due to the reasons discussed above (Reviewer 2, point 1).

In our view, this would constitute the next chapter towards our understanding of this process and is beyond the scope of our current manuscript, which focused on the identification of FimT as a novel DNA receptor for natural transformation and its subsequent structural and biophysical characterisation.

4. Fig. 5b – absence of complementation with FimT orthologs from Pa and Xc could result either because these proteins do not assemble into the legionella T4P, or because they do not natively promote DNA-binding (although the chimeric construct does demonstrate that the C-terminal portion of the Pa FimT is capable of promoting DNA binding when appended to the

end of Lp FimT). Again, this should be acknowledged in the text or tested further as discussed above.

We have improved our discussion of these results in the text (Lines 288-303).

Minor:

Line 129 – “...complete loss of natural transformation...”

Thank you very much. We have made this change (Line 135).

Line 329-330 – the authors should elaborate on why they believe this 2:1 interaction is not likely to be physiologically relevant. Because only a single copy of FimT is hypothesized to be at the pilus tip? Or conformationally even if there were two copies of FimT in the pilus, they would not be able to bind both sides of dsDNA?

We thank the reviewer for this question. Currently we do not know how many copies of FimT are present in the pilus, and whether FimT is incorporated within the pilus fibre or at the tip. Regardless, we think that even if >1 FimT copy is present in the pilus, it would be difficult to envisage how two FimT's could be positioned such that they would engage opposite sides of the dsDNA, in the manner we proposed in our *in vitro* DNA binding experiments. We have improved our discussion in the text (Lines 363-370).

Line 359-360 – change “...on the other hand, both proteins showed near wild-type levels of twitching.” to “...on the other hand, loss of either protein showed near wild-type levels of twitching.”

Thank you very much. We have made this change as suggested (Line 401).

Reviewers' Comments:

Reviewer #1:

Remarks to the Author:

I have read the latest version of this manuscript and I am satisfied with the revisions, including those directly in response to my comments.

Reviewer #2:

Remarks to the Author:

The authors have responded well to all the comments made by the reviewers. The difficulties in unambiguously detecting the presence of a minor pilin, such as FimT, within the pilus fibre are well understood within the field. In their responses, the authors make it clear that they have investigated this point as far as is reasonably possible. Similarly, there are challenges involved in modelling the incorporation of FimT into the fibre, not least because it is unclear whether it is located at the tip or not. I think it is reasonable to conclude that their proposal for the DNA binding site on FimT is at least consistent with current pilus models, but I understand why they do not wish to speculate excessively on this matter.

Reviewer #3:

Remarks to the Author:

The inability to experimentally demonstrate that the fimT mutant still assembles pili is a minor weakness of the revised study. However, the technical issues that the authors mention justify the inability to easily address this question. Also, in the context of the entire body of work (including the new data with fimT point mutants that do not impact DNA binding), there is still very strong evidence to support the proposed model.

Something for the authors to consider moving forward - if surface piliation can be detected using the Δ pilT mutant by Western blot analysis. They can use this strain background to test questions about pilus assembly. For example, if assembly is not affected by deletion of fimT - this should be measurable in the Δ fimT Δ pilT double mutant even if they cannot study this in a background where pilT is intact (since cells make very few pili).

In conclusion, the authors have reasonably addressed the concerns I raised in the initial round of review. This manuscript makes an important and broadly relevant contribution to the field and certainly merits publication in Nature Communications.